# Discriminative Estimation of Total Variation Distance: A Fidelity Auditor for Generative Data

## Abstract

With the proliferation of generative AI and the increasing volume of generative data (also called as synthetic data), assessing the fidelity of generative data has become a critical concern. In this paper, we propose a discriminative approach to estimate the total variation (TV) distance between two distributions as an effective measure of generative data fidelity. Our method quantitatively characterizes the relation between the Bayes risk in classifying two distributions and their TV distance. Therefore, the estimation of total variation distance reduces to that of the Bayes risk. In particular, this paper establishes theoretical results regarding the convergence rate of the estimation error of TV distance between two Gaussian distributions. We demonstrate that, with a specific choice of hypothesis class in classification, a fast convergence rate in estimating the TV distance can be achieved. Specifically, the estimation accuracy of the TV distance is proven to inherently depend on the separation of two Gaussian distributions: smaller estimation errors are achieved when the two Gaussian distributions are farther apart. This phenomenon is also validated empirically through extensive simulations. In the end, we apply this discriminative estimation method to rank fidelity of synthetic image data using the MNIST/CIFAR-10 dataset.

## 1 Introduction

Evaluating the discrepancy between distributions has been a prominent research topic in the statistics and machine learning communities, as evidenced by its extensive applications in hypothesis testing (Gerber et al., 2023; Yang et al., 2018) and generative data evaluation (Sajjadi et al., 2018; Snoke et al., 2018). Particularly in recent years, considerable research efforts have been dedicated to the development of generative models, resulting in a boom in generative data. Within this context, assessing the fidelity of generative data to real data is vital for ensuring the significance of downstream tasks trained on these generative data.

In practice, the fidelity of generative data can be measured via some statistical divergences, such as Kullback-Leibler divergence, Jensen-Shannon divergence, and Total Variation (TV) distance. However, estimating these statistical divergences faces significant hurdles due to the high-dimensional complexity and intricate correlations within the data. These challenges partly explain why the existing frameworks for fidelity evaluation Jordon et al. (2022) predominantly rely on low-dimensional surrogate metrics, such as marginal distributions (Zhang et al., 2014) and correlation plots. To avoid directly computing distributional distances in high dimensions, researchers have proposed several approaches to audit fidelity. These include comparing the density of synthetic and real distributions only over random subsets of datasets (Bowen & Snoke, 2019), or quantifying the similarity between real and synthetic data using precision (quality of synthetic samples) and recall (diversity of synthetic samples) (Sajjadi et al., 2018).

To have a more comprehensive auditing, we realize the necessity and importance of distance estimation at the *distributional* level. To develop an effective approach to estimate the (particularly high dimensional) distributional distance, we start with the TV distance as the metric to compare two distributions, which stands out as the premier metric for evaluating generative data quality in the literature (Tao et al., 2021; Zhang et al., 2014). Our key insight is to frame the TV distance between

two distributions as the Bayes risk in a classification task for distinguishing between them. Thus, the problem of estimating TV distance can be converted into estimating Bayes risk in classification.

We establish theoretical results regarding the convergence rate of the estimation error of TV distance between two Gaussian distributions, which is further extended to the exponential family. Specifically, we show that the proposed estimator converges to the true TV distance in probability at a faster convergence rate compared with results in Rubenstein et al. (2019); Sreekumar & Goldfeld (2022). Interestingly, our theory (one-dimensional Gaussian case (Theorem 3.6)) confirms a phenomenon that the estimation of TV distance inherently depends on the level of separation between two distributions: the farther apart the two distributions are, the easier the estimation task becomes. This phenomenon is validated in extensive simulations (Figure 2). Our theory is developed under the Gaussian assumption that is supported by the normality of generative data embeddings found in images (Kynkäänniemi et al., 2023) and text data (Chun, 2024). In numerical experiments, we utilized our method to compare images generated by generative adversarial networks (GANs; Goodfellow et al., 2020), showing that our method accurately ranks data fidelity based on different types of embeddings (Table 5).

## 1.1 RELATED WORK

There are three related lines of research: the estimation of statistical divergences, the total variation (TV) distance between two Gaussian distributions, and the fidelity evaluation of synthetic data. Below, we provide an overview of relevant studies and highlight how they differ from our own work.

**Statistical divergence estimation.** Contemporary methodologies for estimating divergence metrics predominantly rely on employing plug-in density estimators as surrogates for the densities within these metrics. Moon & Hero (2014) employ a kernel density estimator to estimate the density ratio within the $f$-divergence family. Similarly, Noshad et al. (2017) propose using $k$-nearest neighbor to approximate the continuous density function ratio within the $f$-divergence family. Rubenstein et al. (2019) introduce a random mixture estimator to approximate the $f$-divergence between two probability distributions. Additionally, Sreekumar & Goldfeld (2022) establish non-asymptotic absolute error bounds for the use of neural networks in approximating $f$-divergences. Existing methods primarily nonparametric estimation based, which are hindered by the curse of dimensionality and often overlook the separation between two distributions. Interestingly, our developed method frames the divergence estimation problem as a classification problem that takes into account of the separation gap closely connected with the classic low-noise assumption in classification.

**TV distance between Gaussian distributions.** Devroye et al. (2018) investigate the total variation distance between two high-dimensional Gaussians with the same mean, providing both lower and upper bounds for their total variation distance. Davies et al. (2022) derive new lower bounds on the total variation distance between two-component Gaussian mixtures with a shared covariance matrix by examining the characteristic function of the mixture. Building upon the work of Devroye et al. (2018), Barabesi & Pratelli (2024) improve the results by providing a tighter bound for the total variation distance between two high-dimensional Gaussian distributions based on a more delicate bound for the cumulative distribution function of Gaussians. Existing works on the TV distance between Gaussian distributions primarily focus on deriving upper and lower bounds rather than establishing effective estimation methods based on finite samples.

**Fidelity Evaluation.** To evaluate the fidelity of synthetic data, besides $f$-divergence metrics such as total variation (TV) distance (Zhang et al., 2014) and Kullback-Leibler (KL) divergence (Jiang, 2018), another common metric is the Maximum Mean Discrepancy (MMD) (Sutherland et al., 2016; Li et al., 2017). For instance, Li et al. (2017) directly used MMD as an optimization target to assess the quality of synthetic data. Additionally, in the domain of computer vision, the Fréchet Inception Distance (FID) score (Heusel et al., 2017) is the primary metric used to assess the quality of images generated by generative models. It quantifies the similarity between the distributions of real and generated images, relying on the Fréchet Distance between two multivariate Gaussian distributions (Fréchet, 1957). Kynkäänniemi et al. (2022) study how the use of ImageNet-pretrained Inception features in FID calculations can lead to discrepancies with human judgment. O'Reilly & Asadi (2021) explore the impact of using pre-trained versus randomly initialized weights in the Inception network for FID computation and discuss the reliability and consistency of FID scores.

## 1.2 PRELIMINARIES

For a random variable $X$, we let $\mathbb{E}_X(\cdot)$ denote the expectation taken with respect to the randomness of $X$. For a random sequence $\{X_n\}_{n=1}^{\infty}$, $X_n \xrightarrow{p} X$ indicates that $X_n$ converges to $X$ in probability. We use bold symbols to represent multivariate objects. In binary classification, the objective is to learn a classifier $f : \mathcal{X} \to \{0, 1\}$ for capturing the functional relationship between the feature vector $\boldsymbol{X} \in \mathcal{X}$ and its associated label $Y \in \{0, 1\}$. The performance of $f$ is usually measured by the 0-1 risk as $R(f) = P(f(\boldsymbol{X}) \neq Y)$, where the expectation is taken with respect to the joint distribution of $(\boldsymbol{X}, Y)$. The optimal classifier $f^\star = \operatorname{argmin}_f R(f)$ refers to the Bayes decision rule, which is obtained by minimizing $R(f)$ in a point-wise manner and given as $f^\star(\boldsymbol{X}) = I\left(\eta(\boldsymbol{X}) \geq \frac{1}{2}\right)$, where $\eta(\boldsymbol{X}) = P(Y = 1|\boldsymbol{X})$ and $I(\cdot)$ is the indicator function.

## 2 DISCRIMINATIVE ESTIMATION OF TOTAL VARIATION DISTANCE

In this section, we present an effective classification-based approach to estimate the underlying total variation (TV) distance between two distributions using two sets of their realizations. Our key insight is to conceptualize the total variation distance as a lower bound of the Bayes Risk for a real-synthetic data classifier. By leveraging the duality between total variation distance and Bayes Risk, we establish a lower bound on the total variation distance. This method can serve as a "Fidelity Auditor" for comparing real and synthetic data, and is directly applicable to arbitrary data synthesizers.

### 2.1 FRAMING TOTAL VARIATION DISTANCE AS CLASSIFICATION PROBLEM.

We denote the sets of real data and synthetic data as $\{\boldsymbol{x}_i\}_{i=1}^n$ and $\{\widetilde{\boldsymbol{x}}_i\}_{i=1}^n$, respectively, where $\boldsymbol{x}_i, \widetilde{\boldsymbol{x}}_i \in \mathbb{R}^p$ are $p$-dimensional continuous vectors. Let $\mathbb{P}(\boldsymbol{x})$ and $\mathbb{Q}(\boldsymbol{x})$ denote the density functions of real and synthetic data, respectively. The total variation (TV) distance between $\mathbb{P}(\boldsymbol{x})$ and $\mathbb{Q}(\boldsymbol{x})$ is given as

$$\mathrm{TV}(\mathbb{P}, \mathbb{Q}) = \frac{1}{2} \int_{\mathbb{R}^p} |\mathbb{P}(\boldsymbol{x}) - \mathbb{Q}(\boldsymbol{x})| d\boldsymbol{x}.$$

For the mixed dataset $\mathcal{D} = \{\boldsymbol{x}_i\}_{i=1}^n \cup \{\widetilde{\boldsymbol{x}}_i\}_{i=1}^n$, the underlying density function can be written as

$$\mathbb{D}(\boldsymbol{x}) = \frac{\mathbb{P}(\boldsymbol{x}) + \mathbb{Q}(\boldsymbol{x})}{2}.$$

As elaborated in the work of Nguyen et al. (2009), estimating $f$-divergences can be equivalently transformed to seek the optimal classifier capable of distinguishing real data from synthetic data. Specifically, we set the labels of real and synthetic samples as 1 and 0, respectively. For any sample $\boldsymbol{x}$, the probability of $\boldsymbol{x}$ being real is given as $\eta(\boldsymbol{x}) = \frac{\mathbb{P}(\boldsymbol{x})}{\mathbb{P}(\boldsymbol{x}) + \mathbb{Q}(\boldsymbol{x})}$. Let $f : \mathbb{R}^p \to \{0, 1\}$ be a classifier used to discriminate real and synthetic samples. The expected classification error can be written as

$$R(f) = \mathbb{E}_{\boldsymbol{X}} \left[ I(f(\boldsymbol{X}) = 1) \frac{\mathbb{Q}(\boldsymbol{X})}{\mathbb{P}(\boldsymbol{X}) + \mathbb{Q}(\boldsymbol{X})} + I(f(\boldsymbol{X}) = 0) \frac{\mathbb{P}(\boldsymbol{X})}{\mathbb{P}(\boldsymbol{X}) + \mathbb{Q}(\boldsymbol{X})} \right], \tag{1}$$

where $\boldsymbol{X} \sim \mathbb{D}$. Therefore, the minimal risk $R(f^\star)$ is then given as

$$R(f^\star) = \frac{1}{2} \int_{\mathbb{R}^p} \min\{\mathbb{P}(\boldsymbol{x}), \mathbb{Q}(\boldsymbol{x})\} d\boldsymbol{x} = \frac{1}{2} - \frac{1}{2} \mathrm{TV}(\mathbb{P}, \mathbb{Q}). \tag{2}$$

It is clear from (2) that the estimation of the total variation between $\mathbb{P}$ and $\mathbb{Q}$ is equivalent to that of the Bayes risk $R(f^\star)$ for the task of discriminating between real and synthetic data.

### 2.2 TOTAL VARIATION DISTANCE LOWER BOUND VIA CLASSIFICATION

Given an estimator $\widehat{f}$ of the optimal classifier $f^\star$, we always have

$$R(\widehat{f}) \geq R(f^\star) = \frac{1}{2} - \frac{1}{2} \mathrm{TV}(\mathbb{P}, \mathbb{Q}).$$

This inequality suggests

$$\mathrm{TV}(\mathbb{P}, \mathbb{Q}) \geq 1 - 2R(\widehat{f}) \triangleq \widehat{\mathrm{TV}}(\mathbb{P}, \mathbb{Q}) \tag{3}$$

for any feasible classifier $\widehat{f}$. Therefore, $\widehat{f}$ provides a means to establish a *lower bound* for the total variation distance between the distributions of real and synthetic data distributions. Each specific classifier $\widehat{f}$ yields a lower bound on the *indistinguishability* between $\mathbb{P}$ and $\mathbb{Q}$. Intuitively, if none of classifiers yields a large lower bound, then the synthetic data $\mathbb{Q}$ can be considered similar to the real data $\mathbb{P}$, indicating that their total variation distance is small.

If the chosen classifier $\widehat{f}$ is *consistent* for achieving minimal risk, that is $\mathcal{E}(\widehat{f}) = R(\widehat{f}) - R(f^\star) = 0$, where $\mathcal{E}(\widehat{f})$ is known as the excess risk, then $\widehat{\mathrm{TV}}(\mathbb{P}, \mathbb{Q})$ appears as a consistent estimator of the real total variation $\mathrm{TV}(\mathbb{P}, \mathbb{Q})$, that is

$$\mathcal{E}(\widehat{f}) = R(\widehat{f}) - R(f^\star) \xrightarrow{p} 0 \Leftrightarrow \mathrm{TV}(\mathbb{P}, \mathbb{Q}) - \widehat{\mathrm{TV}}(\mathbb{P}, \mathbb{Q}) \xrightarrow{p} 0.$$

Here the equivalence of these two convergence in probability is supported by the quantitative relation $\mathrm{TV}(\mathbb{P}, \mathbb{Q}) - \widehat{\mathrm{TV}}(\mathbb{P}, \mathbb{Q}) = 2\mathcal{E}(\widehat{f})$. In the literature, there has been various research efforts devoted to establishing the convergence of $\mathcal{E}(\widehat{f})$ (Audibert & Tsybakov, 2007; Bartlett et al., 2006).

## 3 OPTIMAL ESTIMATION OF TOTAL VARIATION DISTANCE

In this section, we present several examples where achieving an optimal classifier is feasible by choosing a proper hypothesis class. For illustration, we primarily examine a scenario where both real and synthetic data are generated from multivariate Gaussian distributions. Subsequently, we offer an extension to encompass the general exponential family. To establish the tightest convergence rate for the empirical fidelity auditor, we adopt the following low noise assumption in the classification literature (Audibert & Tsybakov, 2007; Bartlett et al., 2006).

**Assumption 3.1 (Low-Noise Condition)** *There exist some positive constants $C_0$ and $\gamma$ such that $P(|\eta(\boldsymbol{x}) - 1/2| < t) \leq C_0 t^\gamma$ for any $t > 0$, where $\gamma$ is referred to as the noise exponent.*

Assumption 3.1 characterizes the behavior of the regression function $\eta$ in the vicinity of the level $\eta(\boldsymbol{x}) = 1/2$, which is paramount for convergence of classifiers. Particularly, a larger value of $\gamma$ indicates smaller noise in the labels, resulting in a faster convergence rate to the optimal classifier.

### 3.1 MULTIVARIATE GAUSSIAN DISTRIBUTION

We start with delving into a scenario where both real and synthetic data follow multivariate normal distributions. Our primary aim is to delineate the optimal function class for training an empirical classifier and assess its convergence towards the optimal classifier. This assumption finds particular prevalence in the domain of generative data, owing to the widespread practice of assuming embeddings of generative data to be normally distributed, such as images (Kynkäänniemi et al., 2023) and text data (Chun, 2024).

Specifically, we assume $\mathbb{P}$ and $\mathbb{Q}$ are two different Gaussian density functions parametrized by $(\boldsymbol{\mu}_1, \boldsymbol{\Sigma}_1)$ and $(\boldsymbol{\mu}_2, \boldsymbol{\Sigma}_2)$, respectively. Under this assumption, the underlying distribution of the mixed dataset $\mathcal{D}$ is $\frac{1}{2}N(\boldsymbol{\mu}_1, \boldsymbol{\Sigma}_1) + \frac{1}{2}(\boldsymbol{\mu}_2, \boldsymbol{\Sigma}_2)$.

**Lemma 3.2** *Given that $\mathcal{D} \sim \frac{1}{2}N(\boldsymbol{\mu}_1, \boldsymbol{\Sigma}_1) + \frac{1}{2}N(\boldsymbol{\mu}_2, \boldsymbol{\Sigma}_2)$, the Bayes decision rule (optimal classifier) for determining the true distribution of a given sample $\boldsymbol{x}$ is*

$$f^\star(\boldsymbol{x}) = I\left( \log\left( \frac{\det(\boldsymbol{\Sigma}_2)}{\det(\boldsymbol{\Sigma}_1)} \right) + (\boldsymbol{x} - \boldsymbol{\mu}_2)^T \boldsymbol{\Sigma}_2^{-1}(\boldsymbol{x} - \boldsymbol{\mu}_2) - (\boldsymbol{x} - \boldsymbol{\mu}_1)^T \boldsymbol{\Sigma}_1^{-1}(\boldsymbol{x} - \boldsymbol{\mu}_1) > 0 \right),$$

*where $\det(\cdot)$ denotes the determinant of a matrix.*

Lemma 3.2 specifies the optimal classifier for discriminating between two multivariate Gaussian distributions. However, directly learning $f^\star$ is often computationally infeasible in practical scenarios. As an alternative approach, we consider employing a plug-in classifier, where we aim to estimate $\eta(\boldsymbol{X}) = \frac{\mathbb{P}(\boldsymbol{X})}{\mathbb{P}(\boldsymbol{X}) + \mathbb{Q}(\boldsymbol{X})}$ through the following optimization task:

$$\widehat{\boldsymbol{\beta}} = \arg\min_{\boldsymbol{\beta} \in \mathbb{R}^d} \frac{1}{2n} \sum_{i=1}^n \left\{ \left( 1 - \frac{\exp(\boldsymbol{\beta}^T \psi(\boldsymbol{x}_i))}{1 + \exp(\boldsymbol{\beta}^T \psi(\boldsymbol{x}_i))} \right)^2 + \left( \frac{\exp(\boldsymbol{\beta}^T \psi(\widetilde{\boldsymbol{x}}_i))}{1 + \exp(\boldsymbol{\beta}^T \psi(\widetilde{\boldsymbol{x}}_i))} \right)^2 \right\} + \lambda \|\boldsymbol{\beta}\|_2^2, \quad (4)$$

where $\psi(\boldsymbol{x}) = (1, x_1, \ldots, x_p, x_1^2, x_1 x_2, \ldots, x_{p-1} x_p, x_p^2)$ being a feature transformation of original features $\boldsymbol{x}$ with $d = (p+2)(p+1)/2$.

Next we denote $\mathcal{H} = \left\{ h(\boldsymbol{x}) = \boldsymbol{\beta}^T \psi(\boldsymbol{x}) : \boldsymbol{\beta} \in \mathbb{R}^d \right\}$ and $\widehat{h}(\boldsymbol{x}) = \widehat{\boldsymbol{\beta}}^T \psi(\boldsymbol{x})$. As long as $\widehat{h}$ is obtained, the plug-in classifier can be obtained as

$$\text{Plug-in Classifier:} \quad \widehat{f}(\boldsymbol{x}) = I \left( \frac{\exp(\widehat{h}(\boldsymbol{x}))}{1 + \exp(\widehat{h}(\boldsymbol{x}))} > \frac{1}{2} \right) = I \left( \widehat{h}(\boldsymbol{x}) > 0 \right). \tag{5}$$

Here, $\widehat{f}$ represents an empirical classifier estimated from $\mathcal{D}$, capable of discerning between real and synthetic data originating from two distinct Gaussian distributions.

**Lemma 3.3** *Define $h_\phi^\star = \arg \min_h \mathbb{E} \left[ (\phi(h(\boldsymbol{X})) - Y)^2 \right]$ with $\phi(x) = \frac{1}{1 + \exp(-x)}$. Given that $\boldsymbol{X} \sim \frac{1}{2} N(\boldsymbol{\mu}_1, \boldsymbol{\Sigma}_1) + \frac{1}{2}(\boldsymbol{\mu}_2, \boldsymbol{\Sigma}_2)$ and $P(Y = 1|\boldsymbol{X}) = \frac{\mathbb{P}(\boldsymbol{X})}{\mathbb{P}(\boldsymbol{X}) + \mathbb{Q}(\boldsymbol{X})}$, we have*

$$h_\phi^\star(\boldsymbol{x}) = \log \left( \frac{\det(\boldsymbol{\Sigma}_2)}{\det(\boldsymbol{\Sigma}_1)} \right) + (\boldsymbol{x} - \boldsymbol{\mu}_2)^T \boldsymbol{\Sigma}_2^{-1} (\boldsymbol{x} - \boldsymbol{\mu}_2) - (\boldsymbol{x} - \boldsymbol{\mu}_1)^T \boldsymbol{\Sigma}_1^{-1} (\boldsymbol{x} - \boldsymbol{\mu}_1).$$

Lemma 3.3 validates the effectiveness of (4) in obtaining an empirical classifier. Specifically, as the sample size tends towards infinity, $\widehat{h}$ becomes consistent with $f^\star$ in sign. Therefore, the plug-in classifier $\widehat{f}$ can be used as a surrogate for $f^\star$ to calculate the total variation between $\mathbb{P}$ and $\mathbb{Q}$. To theoretically validate this claim, we demonstrate in Theorem 3.4 that our developed discriminative estimation of the total variation between two Gaussian distributions exhibits a fast convergence rate of $O \left( (d \log(n)/n)^{\frac{\gamma+1}{\gamma+2}} \right)$. This result aligns with the optimal convergence rate in classification under the same assumptions as presented in (Bartlett et al., 2006; Tsybakov, 2004).

Moreover, our theoretical result unveils two intriguing phenomena:

1. When an appropriate function class is chosen for classification, the estimation of the total variation between two Gaussian distributions remains robust against data dimension compared to nonparametric density estimation and neural estimation approaches (Sreekumar & Goldfeld, 2022);
2. The estimation error of total variation inherently depends on the difference between $\mathbb{P}$ and $\mathbb{Q}$, such that a faster convergence rate is achieved when the real total variation distance between $\mathbb{P}$ and $\mathbb{Q}$ is larger (larger values of $\gamma$ or smaller values of $C_0$ in Assumption 3.1).

The second phenomenon is striking because it suggests that the difficulty of estimating total variation diminishes significantly when the true variation is substantial. Despite lacking theoretical validation in existing literature, this result is intuitively comprehensible. In Figure 1, we provide a toy example illustrating that $\mathbb{P}$ and $\mathbb{Q}$ have completely disjoint supports, resulting in a true total variation of one. It can be observed that regardless of the number of samples used to compute the empirical total variation, the estimated total variation is consistent with zero estimation error.

**Theorem 3.4** *If $\mathbb{P}$ and $\mathbb{Q}$ are two different Gaussian density functions parametrized by $(\boldsymbol{\mu}_1, \boldsymbol{\Sigma}_1)$ and $(\boldsymbol{\mu}_2, \boldsymbol{\Sigma}_2)$, respectively. Under Assumption 3.1, we have*

$$\mathbb{E}_{\mathcal{D}} \left\{ \widehat{\text{TV}}(\mathbb{P}, \mathbb{Q}) - \text{TV}(\mathbb{P}, \mathbb{Q}) \right\} \lesssim C_0^{\frac{1}{\gamma+2}} \left( \frac{d \log n}{2n} \right)^{\frac{\gamma+1}{\gamma+2}}, \tag{6}$$

*where $\widehat{\text{TV}}(\mathbb{P}, \mathbb{Q}) = 1 - 2R(\widehat{f})$ with $\widehat{f}$ being the plug-in classifier given by (5) with $\lambda \asymp d \log(n)/n$ and $C_0$ and $\gamma$ are as defined in Assumption 3.1.*

**Lemma 3.5** *Suppose that $\boldsymbol{X} \sim \frac{1}{2} N(\boldsymbol{\mu}_1, \boldsymbol{\Sigma}) + \frac{1}{2} N(\boldsymbol{\mu}_2, \boldsymbol{\Sigma})$, for any $c < 1/2$, we have*

$$P(|\eta(\boldsymbol{X}) - 1/2| < t) \leq \frac{2t}{(1 - 2c)\sqrt{\pi} \|\boldsymbol{\mu}_1 - \boldsymbol{\mu}_2\|_{\boldsymbol{\Sigma}}},$$

*where $\|\boldsymbol{\mu}_1 - \boldsymbol{\mu}_2\|_{\boldsymbol{\Sigma}} = \sqrt{(\boldsymbol{\mu}_1 - \boldsymbol{\mu}_2)^T \boldsymbol{\Sigma}^{-1} (\boldsymbol{\mu}_1 - \boldsymbol{\mu}_2)}$ and $\eta(\boldsymbol{x}) = \frac{\mathbb{P}(\boldsymbol{x})}{\mathbb{P}(\boldsymbol{x}) + \mathbb{Q}(\boldsymbol{x})}$.*

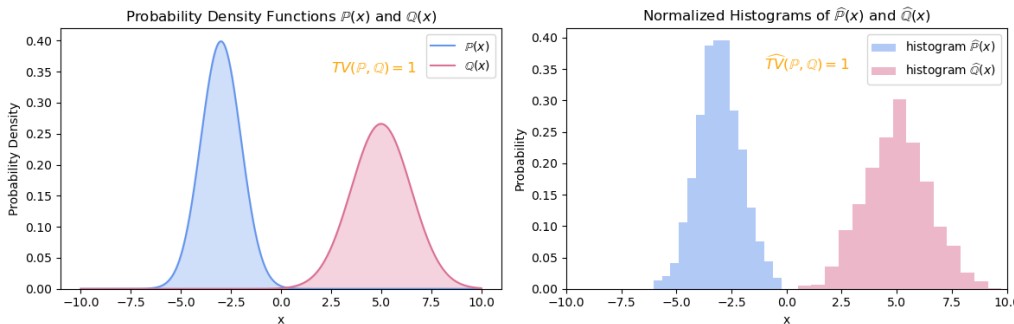

Figure 1: In this case, the supports of $\mathbb{P}$ and $\mathbb{Q}$ are completely non-overlapping, and hence Assumption 3.1 holds with $C_0 = 0$ and any $\gamma > 0$. It is evident that the estimation error in (6) is zero due to the disjoint nature of the histograms for any value of $n$ in this example.

In Lemma 3.5, we verify Assumption 3.1 for the case when $\mathbb{P}$ and $\mathbb{Q}$ are two multivariate Gaussian distributions with identical covariance matrices. This quantifies the values of $C_0$ and $\gamma$, further clarifying the convergence rate developed in (6).

**Theorem 3.6** *Suppose $X \sim \frac{1}{2}N(\boldsymbol{\mu}_1, \boldsymbol{\Sigma}) + \frac{1}{2}N(\boldsymbol{\mu}_2, \boldsymbol{\Sigma})$. With this, (6) becomes*

$$\mathbb{E}_{\mathcal{D}}\left\{\widehat{\mathrm{TV}}(\mathbb{P}, \mathbb{Q}) - \mathrm{TV}(\mathbb{P}, \mathbb{Q})\right\} \lesssim \left(\frac{1}{\|\boldsymbol{\mu}_1 - \boldsymbol{\mu}_2\|_{\boldsymbol{\Sigma}}}\right)^{\frac{1}{3}} \left(\frac{d \log n}{2n}\right)^{\frac{2}{3}},$$

*where $\|\boldsymbol{\mu}_1 - \boldsymbol{\mu}_2\|_{\boldsymbol{\Sigma}} = \sqrt{(\boldsymbol{\mu}_1 - \boldsymbol{\mu}_2)^T \boldsymbol{\Sigma}^{-1}(\boldsymbol{\mu}_1 - \boldsymbol{\mu}_2)}.$*

In Theorem 3.6, we present a detailed analysis of (6) specifically tailored to the Gaussian case with identical covariance matrices. This analysis includes the explicit determination of the constants $C_0$ and $\gamma$ as defined in Assumption 3.1. Specifically, we show that $C_0 \asymp 1/\|\boldsymbol{\mu}_1 - \boldsymbol{\mu}_2\|_{\boldsymbol{\Sigma}}$ and $\gamma = 1$. Our findings demonstrate that the proposed discriminative estimation method achieves a rapid convergence rate of $O\left(\|\boldsymbol{\mu}_1 - \boldsymbol{\mu}_2\|_{\boldsymbol{\Sigma}}^{-1/3} n^{-\frac{2}{3}}\right)$, accompanied by a logarithmic factor. Notably, as $\|\boldsymbol{\mu}_1 - \boldsymbol{\mu}_2\|_{\boldsymbol{\Sigma}}$ tends towards infinity, the convergence rate accelerates, aligning with our second observation mentioned earlier.

## 3.2 EXTENSION TO EXPONENTIAL FAMILY

We extend our Gaussian result to encompass the broader exponential family. Specifically, we address the question of determining the appropriate function class for estimating the total variation between two exponential-type random variables. With the appropriate choice of function classes, similar results for estimating the total variation can be derived, building upon the risk of the resulting classifier.

For any exponential-type random variable $\boldsymbol{X}$, the associated probability density function can typically be expressed in the general form

$$f_{\boldsymbol{X}}(\boldsymbol{x}|\boldsymbol{\theta}) = h(\boldsymbol{x}) \cdot \exp\left[\boldsymbol{\eta}(\boldsymbol{\theta}) \cdot \boldsymbol{T}(\boldsymbol{x}) - A(\boldsymbol{\theta})\right],$$

where $h(\cdot)$, $\boldsymbol{T}(\cdot)$, $\boldsymbol{\eta}(\cdot)$, and $A(\cdot)$ are functions that uniquely depend on the type of $\boldsymbol{X}$.

**Theorem 3.7** *Let $\mathbb{P}(\boldsymbol{x})$ and $\mathbb{Q}(\boldsymbol{x})$ be the density functions of two different random variables from the exponential family:*

$$\mathbb{P}(\boldsymbol{x}) = h_1(\boldsymbol{x}) \cdot \exp\left[\boldsymbol{\eta}_1(\boldsymbol{\theta}_1) \cdot \boldsymbol{T}_1(\boldsymbol{x}) - A_1(\boldsymbol{\theta}_1)\right],$$
$$\mathbb{Q}(\boldsymbol{x}) = h_2(\boldsymbol{x}) \cdot \exp\left[\boldsymbol{\eta}_2(\boldsymbol{\theta}_2) \cdot \boldsymbol{T}_2(\boldsymbol{x}) - A_2(\boldsymbol{\theta}_2)\right].$$

*Then the optimal classifier for minimizing (1) is given as*

$$f^{\star}(x) = I\left(\log\left(\frac{h_1(\boldsymbol{x})}{h_2(\boldsymbol{x})}\right) + A_2(\boldsymbol{\theta}_2) - A_1(\boldsymbol{\theta}_1) + \boldsymbol{T}_1(\boldsymbol{x})\boldsymbol{\eta}_1(\boldsymbol{\theta_1}) - \boldsymbol{T}_2(\boldsymbol{x})\boldsymbol{\eta}(\boldsymbol{\theta_2}) > 0\right). \quad (7)$$

*Furthermore, the total variation between $\mathbb{P}(\boldsymbol{x})$ and $\mathbb{Q}(\boldsymbol{x})$ is given as $\mathrm{TV}(\mathbb{P}, \mathbb{Q}) = 2R(f^{\star}) - 1$.*

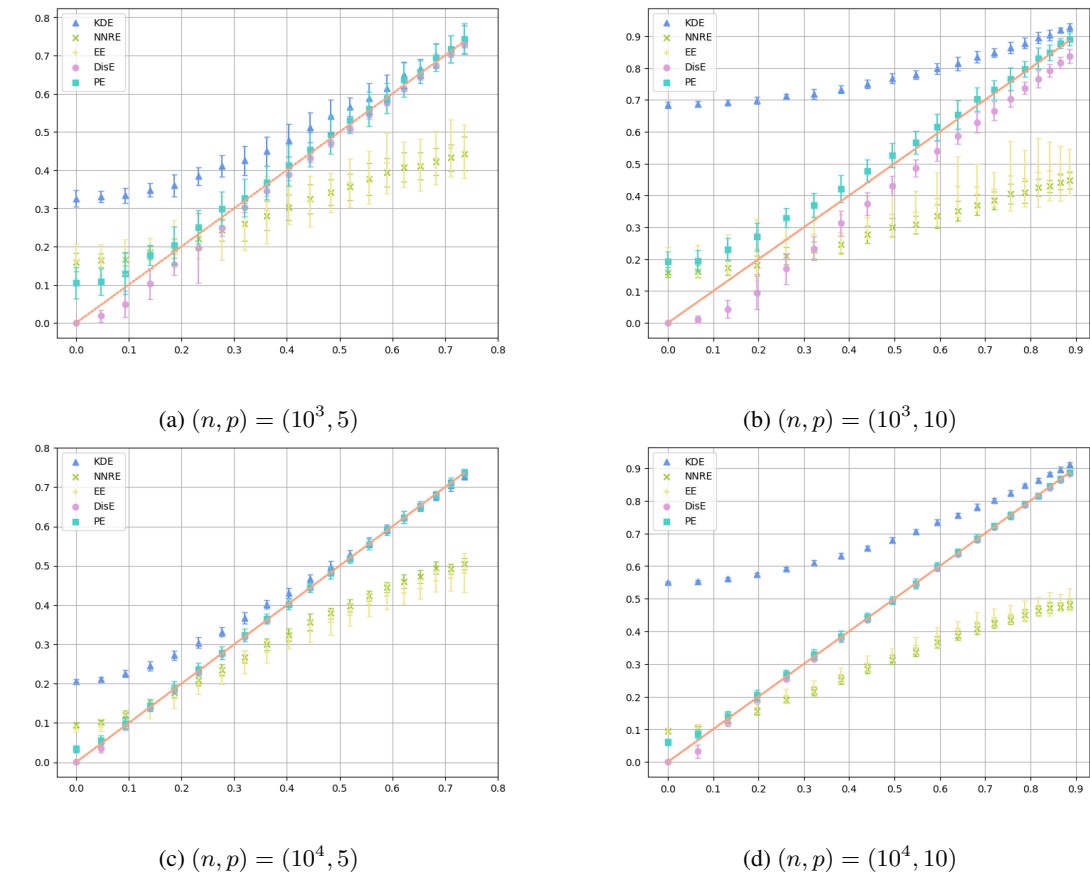

(a) $(n, p) = (10^3, 5)$

(b) $(n, p) = (10^3, 10)$

(c) $(n, p) = (10^4, 5)$

(d) $(n, p) = (10^4, 10)$

Figure 2: True total variation ($x$-axis) versus estimated total variation ($y$-axis) in cases $(n, p) \in \{10^3, 10^4\} \times \{5, 10\}$ under varying disparity between two Gaussian distributions.

Theorem 3.7 elucidates the optimal classifier for discriminating between two random variables from the exponential family, providing a method to calculate the total variation between their underlying distributions. Furthermore, Theorem 3.7 also explicates the appropriate class of margin classifiers when the underlying distributions are from exponential family. For illustration, in the following, we outline the appropriate selection of function classes for different combinations between four exponential-type univariate random variables, as summarized in Table 1. The extension to other exponential-type random variables and multivariate cases can be derived analytically.

Table 1: The choice of function class takes the form as $\mathcal{H} = \{f(\boldsymbol{x}) = \boldsymbol{\beta}^T \psi(\boldsymbol{x}) : \boldsymbol{\beta} \in \mathbb{R}^d\}$. Below presents the explicit form of $\psi(\boldsymbol{x})$ under different combinations of types of $\mathbb{P}$ and $\mathbb{Q}$. Due to the symmetry between $\mathbb{P}$ and $\mathbb{Q}$, we display only the upper triangular results in this table.

| $\mathbb{Q}$ \ $\mathbb{P}$ | Gaussian | Exponential | Gamma | Beta |
|---|---|---|---|---|
| Gaussian | $(1, x, x^2)$ | $(1, x, x^2)$ | $(1, x, x^2, \log x)$ | $(1, x, x^2, \log x, \log(1-x))$ |
| Exponential | - | $(1, x)$ | $(1, x, \log x)$ | $(1, x, \log x, \log(1-x))$ |
| Gamma | - | - | $(1, x, \log x)$ | $(1, x, \log x, \log(1-x))$ |
| Beta | - | - | - | $(1, \log x, \log(1-x))$ |

## 4 EXPERIMENTS

In this section, we showcase the superior performance of the developed discriminative method (DisE) for estimating the total variation between two Gaussian distributions. For each simulated setting,

we report the average results for all simulation settings, accompanied by their respective standard deviations calculated over 20 replications, presented in parentheses.

**Comparison Methods and Evaluation Metrics.** Existing methods for estimating divergence metrics predominantly rely on a plug-in estimation approach, typically applied to either two separate density functions or their density ratio. In this experiment, we consider kernel density estimation (KDE; (Sasaki et al., 2015)) for the former type of estimator. For the latter, we explore two nearest neighbor type estimators, including the ensemble estimation (EE; (Moon & Hero, 2014)) and nearest neighbor ratio estimation (NNRE; (Noshad et al., 2017)). Furthermore, we incorporate a parameter estimation (PE) approach, which entails approximating the total variation through the Monte Carlo method based on sample mean and covariance matrix. As a baseline, we utilize the Monte Carlo method to calculate the true total variation based on true means and covariance matrices. The performance of all methods are evaluated in three aspects, including robustness, computational time, and estimation error measured in absolute error.

**Experimental Setting.** We conduct a comprehensive analysis of the impact of sample size and data dimension on the performance of various estimators. Specifically, we consider $\mathbb{P}$ as a Gaussian distribution with mean $\boldsymbol{\mu}_1 = \mathbf{0}_p$ and covariance matrix $\boldsymbol{\Sigma}_1 = \boldsymbol{I}_{p \times p}$. In contrast, $\mathbb{Q}$ is a Gaussian distribution with mean $\boldsymbol{\mu}_2$ uniformly generated from $[0, 1]^p$ and covariance matrix $\boldsymbol{\Sigma}_2 = \boldsymbol{I}_{p \times p} + \boldsymbol{E}$, where $\boldsymbol{E}$ is a symmetric noise matrix. We compare the performance of our proposed method with that of existing estimation methods across different data dimensions, sample sizes, and differences between the means of two distributions. For each fixed setting, we conduct 20 replications to calculate the standard deviations, which serve as a measure of the robustness of the estimation accuracy.

**Experimental Result.** Figure 2 shows that the DisE and PE methods provide the most accurate estimates of the true total variation distance across all scenarios. The KDE approach tends to overestimate the total variation in cases of smaller disparity, while the NNRE and EE approaches tend to underestimate it in cases of larger disparity. Notably, as the true total variation increases, the accuracy of our proposed DisE method improves, which aligns perfectly with the theoretical results established in Theorem 3.4. Furthermore, compared to other methods, our proposed method is less sensitive to data dimensionality.

**Robustness Study.** To further validate the robustness of our proposed method, we repeatedly compare the estimation results across different dimensions ranging from 2 to 12, and examine the estimation results under different levels of noise added to data. The average estimation errors under varying disparities between two distributions are reported in Figure 3 and Table 2. Clearly, both DisE and PE consistently exhibit smaller estimation errors, while the other approaches show increasing errors as the dimension expands. Table 2 demonstrates that the DisE approach achieves higher accuracy and lower variance compared to the PE approach. Figure 4 and Table 3 show the average estimation errors under varying levels of variances of noise added to data. The estimation errors of all approaches show a growing pattern with the increase of noise level, and the proposed DisE approach has a relatively lower estimation error compared with other methods. Overall, these findings confirm the superior robustness and accuracy of the DisE approach in estimating total variation distance under varying dimensions and noise levels.

**Exponential Family.** We extended the simulation experiment to Exponential family to examine the performance of our proposed DisE approach. Table 4 show the average estimation errors and standard deviations of total variation estimation of all methods for Exponential distribution and Gamma distribution respectively. Both tables demonstrate that DisE approach provides more accurate estimation of total variation with smaller standard deviation.

## 5 REAL APPLICATION - CONSISTENT FIDELITY COMPARISON OF GENERATIVE DATA.

**Experimental Setting.** We evaluate the effectiveness of the DisE, PE, and KDE methods in measuring the fidelity of synthetic data. Using the MNIST dataset (LeCun, 1998) and CIFAR-10 (Krizhevsky et al., 2009) dataset , we train GANs for 100, 300, and 500 epochs, subsequently generating images with each of these models, as illustrated in Figure 5. Due to the high dimensionality and sparsity of image data, we employ pretrained ResNet18 (He et al., 2016) to obtain embeddings of both real and synthetic images. Following the literature, which commonly assumes the normality of embeddings

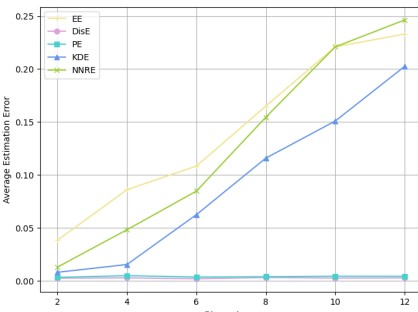

| Method | dim = 2 | dim = 4 | dim = 6 |
|---|---|---|---|
| DisE | **0.002**(0.002) | **0.003**(0.002) | **0.002**(0.001) |
| PE | 0.003(0.002) | 0.005(0.004) | 0.004(0.003) |
| KDE | 0.008(0.005) | 0.015(0.017) | 0.062(0.045) |
| NNRE | 0.013(0.015) | 0.048(0.027) | 0.085(0.059) |
| EE | 0.038(0.019) | 0.086(0.052) | 0.108(0.074) |
| Method | dim = 8 | dim = 10 | dim = 12 |
| DisE | **0.003**(0.002) | **0.002** (0.002) | **0.003**(0.002) |
| PE | 0.004(0.004) | 0.004(0.003) | 0.004(0.003) |
| KDE | 0.115(0.089) | 0.151(0.121) | 0.202(0.154) |
| NNRE | 0.154(0.091) | 0.221(0.118) | 0.246(0.124) |
| EE | 0.165(0.099) | 0.221(0.125) | 0.233(0.124) |

Figure 3: The robustness of estimation errors of all methods with respect to data dimensionality.

Table 2: The averaged estimation errors (standard deviations) of total variation estimation of all methods across various data dimensions.

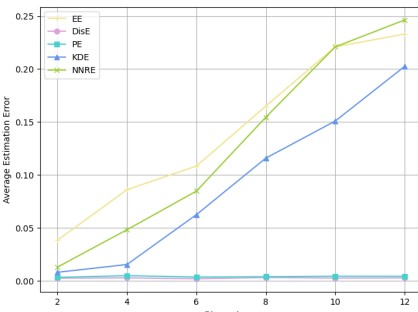

| Method | noise = 0.1 | noise = 0.5 | noise = 1.0 |
|---|---|---|---|
| DisE | **0.003**(0.002) | **0.032**(0.027) | **0.159**(0.117) |
| PE | 0.005(0.004) | 0.033(0.029) | 0.176(0.120) |
| KDE | 0.052(0.043) | 0.074(0.060) | 0.203(0.131) |
| NNRE | 0.054(0.041) | 0.054(0.035) | 0.129(0.100) |
| EE | 0.079(0.063) | 0.069(0.051) | 0.129(0.092) |
| Method | noise = 1.5 | noise = 2.0 | noise = 2.5 |
| DisE | **0.310**(0.174) | **0.423** (0.207) | **0.494**(0.225) |
| PE | 0.350(0.169) | 0.478(0.187) | 0.557(0.195) |
| KDE | 0.376(0.173) | 0.501(0.189) | 0.577(0.196) |
| NNRE | 0.294(0.153) | 0.437(0.179) | 0.524(0.198) |
| EE | 0.294(0.149) | 0.452(0.179) | 0.569(0.200) |

Figure 4: The robustness of estimation errors of all methods with respect to noise added to data (dimension = 5).

Table 3: The averaged estimation errors (standard deviations) of total variation estimation of all methods across different noise variances.

of generative data (Kynkäänniemi et al., 2023; Chun, 2024), we then estimate the total variation between each generated dataset and the original MNIST/CIFAR-10 dataset using the DisE, PE, and KDE methods. As illustrated in Figure 5, GANs trained for more epochs generate images of greater fidelity. Consequently, the total variation between real images and synthetic images generated after 100, 300, and 500 epochs should follow a decreasing pattern. Hence, in this experiment, we aim to consistently compare all methods in terms of their ability to provide a correct ranking of fidelity. **Experimental Result.** In Table 5, we present the fidelity of images generated by GANs trained over varying epochs, measured using total variation distance estimated by three methods. The total variation distance between the embeddings of real images and synthetic images generated after 100,

Table 4: The averaged estimation errors (standard deviations) of total variation estimation of all methods for Exponential and Gamma distribution (dimension = 1).

| | Method | True TV = 0 | True TV = 0.30 | True TV = 0.70 | True TV = 0.82 |
|---|---|---|---|---|---|
| | DisE | **0.001**(0.001) | **0.000**(0.001) | **0.000**(0.001) | **0.001**(0.001) |
| | PE | 0.006(0.004) | 0.005(0.005) | 0.003(0.004) | 0.002(0.001) |
| Exponential | KDE | 0.020(0.007) | 0.037(0.007) | 0.053(0.007) | 0.048(0.003) |
| | NNRE | 0.094(0.004) | 0.021(0.009) | 0.002(0.005) | 0.011(0.007) |
| | EE | 0.079(0.008) | 0.015(0.016) | 0.002(0.008) | 0.003(0.022) |
| | Method | True TV = 0 | True TV = 0.25 | True TV = 0.72 | True TV = 0.97 |
| | DisE | **0.001**(0.001) | **0.001**(0.001) | **0.000**(0.001) | **0.000**(0.001) |
| | PE | 0.013(0.008) | 0.001(0.008) | 0.000(0.005) | 0.001(0.002) |
| Gamma | KDE | 0.021(0.005) | 0.001(0.008) | 0.006(0.007) | 0.007(0.003) |
| | NNRE | 0.097(0.003) | 0.016(0.009) | 0.104(0.013) | 0.418(0.010) |
| | EE | 0.081(0.008) | 0.006(0.011) | 0.089(0.021) | 0.440(0.017) |

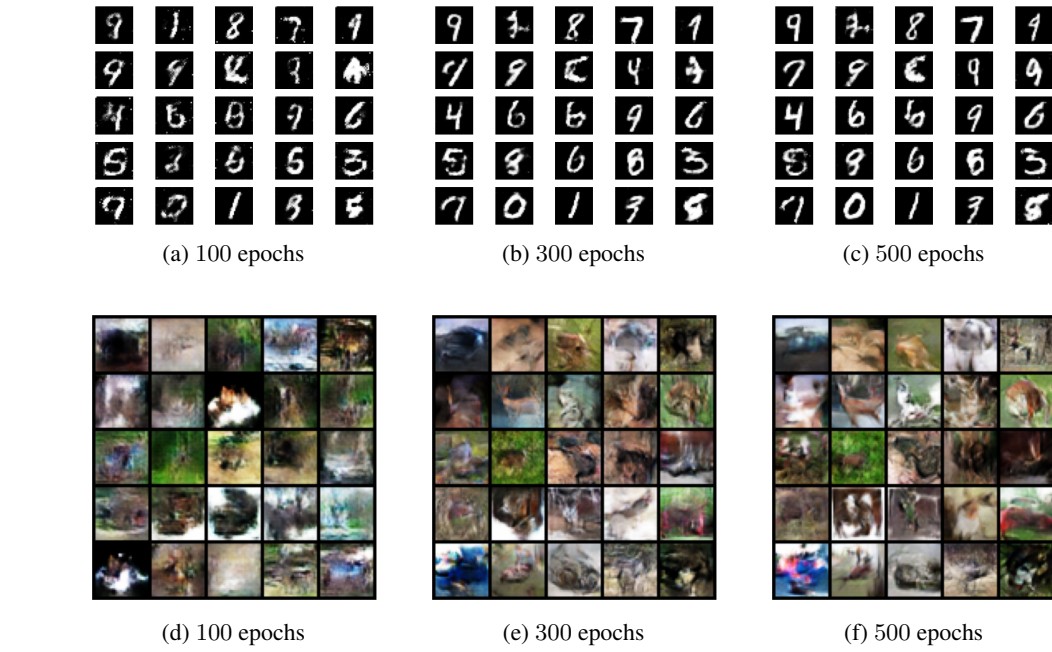

(a) 100 epochs     (b) 300 epochs     (c) 500 epochs

(d) 100 epochs     (e) 300 epochs     (f) 500 epochs

Figure 5: 25 synthetic images generated by GANs after 100, 300, and 500 epochs of training via MNIST/CIFAR-10 dataset are displayed from left to right.

300, and 500 epochs estimated by DisE approach presents a decreasing pattern across all cases, aligning with the expected quality ranking of the generated models. However, the fidelity measured by the PE approach deviates from the expected ranking when the embedding dimension is 50 for the MNIST dataset. Similarly, the fidelity measured by the KDE approach fails to align with the correct ranking when the embedding dimension is 35 for the MNIST dataset and 35 and 50 for the CIFAR-10 dataset. This study demonstrates the effectiveness of the proposed DisE method in measuring the fidelity of synthetic data, providing a correct ranking of quality of generative data.

Table 5: Fidelity rankings of images generated by GANs trained after varying epochs: Fidelity is measured using the total variation estimated by different methods. The dimension of embeddings is set to 20, 35, and 50 for ResNet18.

| Dataset | Method | Embedding-dim | 100 epochs | 300 epochs | 500 epochs | Correct Ranking |
|---------|--------|---------------|------------|------------|------------|-----------------|
| MNIST | DisE | Resnet18-20 | 0.342 (0.068) | 0.153 (0.038) | 0.148 (0.055) | ✓ |
| | | Resnet18-35 | 0.412 (0.074) | 0.187(0.059) | 0.146 (0.050) | ✓ |
| | | Resnet18-50 | 0.436 (0.074) | 0.193 (0.072) | 0.186 (0.041) | ✓ |
| | PE | Resnet18-20 | 0.483 (0.073) | 0.301 (0.051) | 0.286 (0.063) | ✓ |
| | | Resnet18-35 | 0.627 (0.076) | 0.436 (0.065) | 0.431 (0.087) | ✓ |
| | | Resnet18-50 | 0.767 (0.044) | 0.561 (0.061) | 0.563 (0.077) | ✗ |
| | KDE | Resnet18-20 | 0.768 (0.025) | 0.707 (0.017) | 0.703 (0.026) | ✓ |
| | | Resnet18-35 | 0.907 (0.014) | 0.871 (0.013) | 0.872 (0.020) | ✗ |
| | | Resnet18-50 | 0.967 (0.005) | 0.944 (0.007) | 0.943 (0.010) | ✓ |
| CIFAR10 | DisE | Resnet18-20 | 0.332(0.031) | 0.274(0.035) | 0.255(0.042) | ✓ |
| | | Resnet18-35 | 0.463(0.038) | 0.378(0.055) | 0.348(0.055) | ✓ |
| | | Resnet18-50 | 0.577(0.041) | 0.483(0.059) | 0.444(0.038) | ✓ |
| | PE | Resnet18-20 | 0.366(0.027) | 0.309(0.032) | 0.291(0.027) | ✓ |
| | | Resnet18-35 | 0.532(0.032) | 0.462(0.033) | 0.437(0.032) | ✓ |
| | | Resnet18-50 | 0.682(0.029) | 0.604(0.031) | 0.572(0.031) | ✓ |
| | KDE | Resnet18-20 | 0.899(0.004) | 0.893(0.003) | 0.891 (0.004) | ✓ |
| | | Resnet18-35 | 0.990(0.001) | 0.990(0.001) | 0.989(0.001) | ✗ |
| | | Resnet18-50 | 0.999(0.001) | 0.999(0.001) | 0.999(0.001) | ✗ |

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



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

## A DISCUSSION

In this paper, we propose a novel approach to estimate the TV distance between two distributions using a classification-based method. This method leverages the quantitative relationship between Bayes risk and TV distance. Specifically, we examine a scenario where both distributions are Gaussian, establishing theoretical results regarding the convergence of our approach. Our findings reveal an intriguing phenomenon: the estimation error of the TV distance is dependent on the true separation between the distributions. In other words, the TV distance is easier to estimate when the distributions are farther apart. The experimental results demonstrate the superior performance of our proposed discriminative estimation approach over several existing methods in estimating total variation distance. While currently confined to this particular metric, our discriminative approach holds promise for broader applications in estimating various divergence metrics. Future endeavors will focus on extending our method to encompass other divergence metrics and establishing statistical assurances for estimation accuracy.

## B PROOF OF LEMMAS

### B.1 PROOF OF LEMMA 3.5

**Proof.** Given that $\boldsymbol{\Sigma}_1 = \boldsymbol{\Sigma}_2 = \boldsymbol{\Sigma}$ and $t \in (0, 1/2)$, we have

$$\left| \frac{\mathbb{P}(\boldsymbol{X})}{\mathbb{P}(\boldsymbol{X}) + \mathbb{Q}(\boldsymbol{X})} - \frac{1}{2} \right| < t \Leftrightarrow \log\left( \frac{1-2t}{1+2t} \right) < \log\left( \frac{\mathbb{P}(\boldsymbol{X})}{\mathbb{Q}(\boldsymbol{X})} \right) < \log\left( \frac{1+2t}{1-2t} \right).$$

Plugging the densities of $\mathbb{P}(\boldsymbol{X})$ and $\mathbb{Q}(\boldsymbol{X})$ into the above formula yields that

$$\log\left( \frac{\mathbb{P}(\boldsymbol{X})}{\mathbb{Q}(\boldsymbol{X})} \right) = (\boldsymbol{x} - \boldsymbol{\mu}_2)^T \boldsymbol{\Sigma}_2^{-1}(\boldsymbol{x} - \boldsymbol{\mu}_2) - (\boldsymbol{x} - \boldsymbol{\mu}_1)^T \boldsymbol{\Sigma}_1^{-1}(\boldsymbol{x} - \boldsymbol{\mu}_1)$$

$$= 2(\boldsymbol{\mu}_1 - \boldsymbol{\mu}_2)^T \boldsymbol{\Sigma}^{-1} \boldsymbol{x} + \boldsymbol{\mu}_2^T \boldsymbol{\Sigma}^{-1} \boldsymbol{\mu}_2 - \boldsymbol{\mu}_1^T \boldsymbol{\Sigma}^{-1} \boldsymbol{\mu}_1$$

$$= 2(\boldsymbol{\mu}_1 - \boldsymbol{\mu}_2)^T \boldsymbol{\Sigma}^{-1}(\boldsymbol{x} - \boldsymbol{\mu}_1) + (\boldsymbol{\mu}_2 - \boldsymbol{\mu}_1)^T \boldsymbol{\Sigma}^{-1}(\boldsymbol{\mu}_2 - \boldsymbol{\mu}_1)$$

$$= 2(\boldsymbol{\mu}_1 - \boldsymbol{\mu}_2)^T \boldsymbol{\Sigma}^{-1}(\boldsymbol{x} - \boldsymbol{\mu}_2) - (\boldsymbol{\mu}_2 - \boldsymbol{\mu}_1)^T \boldsymbol{\Sigma}^{-1}(\boldsymbol{\mu}_2 - \boldsymbol{\mu}_1).$$

For ease of notation, we define $\| \cdot \|_{\boldsymbol{\Sigma}}^2$ as

$$\|\boldsymbol{x}\|_{\boldsymbol{\Sigma}}^2 = \boldsymbol{x}^T \boldsymbol{\Sigma}^{-1} \boldsymbol{x}.$$

Moreover, we let $J(t) = \log\left( \frac{1+2t}{1-2t} \right)$ and define

$$\phi_1(\boldsymbol{x}) = \frac{1}{\sqrt{(2\pi)^p \det(\boldsymbol{\Sigma})}} \exp\left\{ -\frac{1}{2} \|\boldsymbol{x} - \boldsymbol{\mu}_1\|_{\boldsymbol{\Sigma}}^2 \right\},$$

$$\phi_2(\boldsymbol{x}) = \frac{1}{\sqrt{(2\pi)^p \det(\boldsymbol{\Sigma})}} \exp\left\{ -\frac{1}{2} \|\boldsymbol{x} - \boldsymbol{\mu}_2\|_{\boldsymbol{\Sigma}}^2 \right\},$$

denote the probability density functions of $N(\boldsymbol{\mu}_1, \boldsymbol{\Sigma})$ and $N(\boldsymbol{\mu}_2, \boldsymbol{\Sigma})$, respectively. Then the probability density function of $\boldsymbol{X}$ is given as $\phi(\boldsymbol{x}) = \frac{1}{2}\phi_1(\boldsymbol{x}) + \frac{1}{2}\phi_2(\boldsymbol{x})$. Define the event $S(t)$ as

$$S(t) = \left\{ \boldsymbol{x} \in \mathbb{R}^p : -J(t) \leq 2(\boldsymbol{\mu}_1 - \boldsymbol{\mu}_2)^T \boldsymbol{\Sigma}^{-1} \boldsymbol{x} + \|\boldsymbol{\mu}_2\|_{\boldsymbol{\Sigma}}^2 - \|\boldsymbol{\mu}_1\|_{\boldsymbol{\Sigma}}^2 \leq J(t) \right\}.$$

Next, we turn to bound $\int_{\boldsymbol{x} \in S(t)} \phi_1(\boldsymbol{x}) d\boldsymbol{x}$. Note that $S(t)$ can be equivalently represented as

$$S(t) = \left\{ \boldsymbol{x} \in \mathbb{R}^p : -J(t) \leq 2(\boldsymbol{\mu}_1 - \boldsymbol{\mu}_2)^T \boldsymbol{\Sigma}^{-1}(\boldsymbol{x} - \boldsymbol{\mu}_1) + \|\boldsymbol{\mu}_2 - \boldsymbol{\mu}_1\|_{\boldsymbol{\Sigma}}^2 \leq J(t) \right\}.$$

Denote that $Y = 2(\boldsymbol{\mu}_1 - \boldsymbol{\mu}_2)^T \boldsymbol{\Sigma}^{-1}(\boldsymbol{x} - \boldsymbol{\mu}_1)$. Clearly, $Y$ follows a normal distribution with mean 0 and variance $4(\boldsymbol{\mu}_1 - \boldsymbol{\mu}_2)^T \boldsymbol{\Sigma}^{-1}(\boldsymbol{\mu}_1 - \boldsymbol{\mu}_2)$. Therefore,

$$\int_{\boldsymbol{x} \in S(t)} \phi_1(\boldsymbol{x}) d\boldsymbol{x} = \int_{-J(t) - \|\boldsymbol{\mu}_2 - \boldsymbol{\mu}_1\|_{\boldsymbol{\Sigma}}^2}^{J(t) - \|\boldsymbol{\mu}_2 - \boldsymbol{\mu}_1\|_{\boldsymbol{\Sigma}}^2} \frac{1}{\sqrt{2\pi}\sigma} e^{-\frac{y^2}{2\sigma^2}} dy$$

$$\leq \int_{-J(t)}^{J(t)} \frac{1}{\sqrt{2\pi}\sigma} e^{-\frac{y^2}{2\sigma^2}} dy = \frac{J(t)}{\sqrt{2\pi} \|\boldsymbol{\mu}_1 - \boldsymbol{\mu}_2\|_{\boldsymbol{\Sigma}}},$$

where $\sigma = 2\sqrt{(\boldsymbol{\mu}_1 - \boldsymbol{\mu}_2)^T \boldsymbol{\Sigma}^{-1} (\boldsymbol{\mu}_1 - \boldsymbol{\mu}_2)} = 2\|\boldsymbol{\mu}_1 - \boldsymbol{\mu}_2\|_{\boldsymbol{\Sigma}}$. Similarly, we have

$$\int_{\boldsymbol{x} \in S(t)} \phi_2(\boldsymbol{x}) d\boldsymbol{x} \leq \frac{J(t)}{\sqrt{2\pi}\|\boldsymbol{\mu}_1 - \boldsymbol{\mu}_2\|_{\boldsymbol{\Sigma}}}.$$

Then we have

$$P(\boldsymbol{X} \in S(t)) = \frac{1}{2}\int_{\boldsymbol{x} \in S(t)} \phi_1(\boldsymbol{x}) d\boldsymbol{x} + \frac{1}{2}\int_{\boldsymbol{x} \in S(t)} \phi_2(\boldsymbol{x}) d\boldsymbol{x} \leq \frac{J(t)}{\sqrt{2\pi}\|\boldsymbol{\mu}_1 - \boldsymbol{\mu}_2\|_{\boldsymbol{\Sigma}}}.$$

Note that $\log\left(\frac{1+2t}{1-2t}\right) \leq \frac{4t}{1-2t}$ for any $t \in [0, 1/2)$. Therefore, we have

$$P(\boldsymbol{X} \in S(t)) \leq \frac{2t}{(1 - 2c)\sqrt{\pi}\|\boldsymbol{\mu}_1 - \boldsymbol{\mu}_2\|_{\boldsymbol{\Sigma}}}, \tag{8}$$

for any $t \in (0, c]$ with $c < 1/2$. This completes the proof.

### B.2 PROOF OF LEMMA 3.2

Given that $\mathcal{D}$ is the mixture of two Gaussian distribution $\mathbb{P}(\boldsymbol{x})$ and $\mathbb{Q}(\boldsymbol{x})$, where

$$\mathbb{P}(\boldsymbol{x}) = (2\pi)^{-\frac{p}{2}} \det(\boldsymbol{\Sigma}_1)^{-\frac{1}{2}} \exp\left(-\frac{1}{2}(\boldsymbol{x} - \boldsymbol{\mu}_1)^T \boldsymbol{\Sigma}_1^{-1} (\boldsymbol{x} - \boldsymbol{\mu}_1)\right),$$

$$\mathbb{Q}(\boldsymbol{x}) = (2\pi)^{-\frac{p}{2}} \det(\boldsymbol{\Sigma}_2)^{-\frac{1}{2}} \exp\left(-\frac{1}{2}(\boldsymbol{x} - \boldsymbol{\mu}_2)^T \boldsymbol{\Sigma}_2^{-1} (\boldsymbol{x} - \boldsymbol{\mu}_2)\right),$$

and $\mathbb{D}(\boldsymbol{x}) = \frac{\mathbb{P}(\boldsymbol{x}) + \mathbb{Q}(\boldsymbol{x})}{2}$. For a classifier $f : \mathbb{R}^p \to \{0, 1\}$, its risk is given as

$$R(f) = \int_{\mathbb{R}^p} \mathbb{D}(\boldsymbol{x}) \big[P\left(Y = 1|\boldsymbol{X}\right) \cdot I(f(\boldsymbol{x}) = 0) + P\left(Y = 0|\boldsymbol{X}\right) \cdot I(f(\boldsymbol{x}) = 1)\big] d\boldsymbol{x}.$$

The term $P\left(Y = 1|\boldsymbol{X}\right) = \frac{\mathbb{P}(\boldsymbol{X})}{\mathbb{P}(\boldsymbol{X}) + \mathbb{Q}(\boldsymbol{X})}$ and $P\left(Y = 0|\boldsymbol{X}\right) = \frac{\mathbb{Q}(\boldsymbol{X})}{\mathbb{P}(\boldsymbol{X}) + \mathbb{Q}(\boldsymbol{X})}$, thus we have

$$R(f) = \int_{\boldsymbol{X}} \mathbb{D}(\boldsymbol{x}) \left[\eta(\boldsymbol{x}) \cdot I(f(\boldsymbol{x}) = 0) + (1 - \eta(\boldsymbol{x})) \cdot I(f(\boldsymbol{x}) = 1)\right] d\boldsymbol{x}.$$

To minimize the risk, the optimal classifier is

$$f^{\star}(\boldsymbol{x}) = I\left(\eta(\boldsymbol{x}) > \frac{1}{2}\right) = I\big(\mathbb{P}(\boldsymbol{x}) > \mathbb{Q}(\boldsymbol{x})\big) = I\left(\log \frac{\mathbb{P}(\boldsymbol{x})}{\mathbb{Q}(\boldsymbol{x})} > 0\right)$$

Next,

$$\begin{aligned}
\frac{\mathbb{P}(\boldsymbol{x})}{\mathbb{Q}(\boldsymbol{x})} &= \frac{\det(\boldsymbol{\Sigma}_1)^{-\frac{1}{2}} \exp\left(-\frac{1}{2}(\boldsymbol{x} - \boldsymbol{\mu}_1)^T \boldsymbol{\Sigma}_1^{-1} (\boldsymbol{x} - \boldsymbol{\mu}_1)\right)}{\det(\boldsymbol{\Sigma}_2)^{-\frac{1}{2}} \exp\left(-\frac{1}{2}(\boldsymbol{x} - \boldsymbol{\mu}_2)^T \boldsymbol{\Sigma}_2^{-1} (\boldsymbol{x} - \boldsymbol{\mu}_2)\right)} \\
&= \frac{\det(\boldsymbol{\Sigma}_2)}{\det(\boldsymbol{\Sigma}_1)} \cdot \exp\left(\frac{1}{2}(\boldsymbol{x} - \boldsymbol{\mu}_2)^T \boldsymbol{\Sigma}_2^{-1} (\boldsymbol{x} - \boldsymbol{\mu}_2) - \frac{1}{2}(\boldsymbol{x} - \boldsymbol{\mu}_1)^T \boldsymbol{\Sigma}_1^{-1} (\boldsymbol{x} - \boldsymbol{\mu}_1)\right).
\end{aligned}$$

Considering that $\text{sign}(\mathbb{P}(\boldsymbol{x}) - \mathbb{Q}(\boldsymbol{x})) = \text{sign}(\log \mathbb{P}(\boldsymbol{x}) - \log \mathbb{Q}(\boldsymbol{x}))$. Therefore, the Bayes classifier can be written as

$$f^{\star}(\boldsymbol{x}) = I\left(\log\left(\frac{\det(\boldsymbol{\Sigma}_2)}{\det(\boldsymbol{\Sigma}_1)}\right) + (\boldsymbol{x} - \boldsymbol{\mu}_2)^T \boldsymbol{\Sigma}_2^{-1} (\boldsymbol{x} - \boldsymbol{\mu}_2) - (\boldsymbol{x} - \boldsymbol{\mu}_1)^T \boldsymbol{\Sigma}_1^{-1} (\boldsymbol{x} - \boldsymbol{\mu}_1) > 0\right).$$

This completes the proof.

### B.3 PROOF OF LEMMA 3.3

We first define $R_\phi(h) = \mathbb{E}\left[(\phi(h(\boldsymbol{X})) - Y)^2\right]$, which can be expressed as

$$\mathbb{E}\left[(\phi(h(\boldsymbol{X})) - Y)^2\right] = \int_{\mathbb{R}^p} \mathbb{D}(\boldsymbol{x})\left[\eta(\boldsymbol{x})(\phi(h(\boldsymbol{x})) - 1)^2 + (1 - \eta(\boldsymbol{x}))\phi^2(h(\boldsymbol{x}))\right]d\boldsymbol{x}.$$

Here $\phi(x) = 1/(1 + \exp(-x))$. For each $\boldsymbol{x}$, we have

$$\eta(\boldsymbol{x})(\phi(h(\boldsymbol{x})) - 1)^2 + (1 - \eta(\boldsymbol{x}))\phi^2(h(\boldsymbol{x}))$$
$$= \eta(\boldsymbol{x})\left(\frac{1}{1 + \exp(h(\boldsymbol{x}))}\right)^2 + (1 - \eta(\boldsymbol{x}))\left(\frac{\exp(h(\boldsymbol{x}))}{1 + \exp(h(\boldsymbol{x}))}\right)^2. \tag{9}$$

Clearly, (9) is minimized when $\phi(h(\boldsymbol{x})) = \eta(\boldsymbol{x})$, leading to

$$h_\phi^\star(\boldsymbol{x}) = \log\left(\frac{\eta(\boldsymbol{x})}{1 - \eta(\boldsymbol{x})}\right) = \log\left(\frac{\mathbb{P}(\boldsymbol{x})}{\mathbb{Q}(\boldsymbol{x})}\right).$$

Finally, we have

$$h_\phi^\star(\boldsymbol{x}) = \log\left(\frac{\mathbb{P}(\boldsymbol{x})}{\mathbb{Q}(\boldsymbol{x})}\right) = \log\left(\frac{\det(\boldsymbol{\Sigma}_2)}{\det(\boldsymbol{\Sigma}_1)}\right) + (\boldsymbol{x} - \boldsymbol{\mu}_2)^T\boldsymbol{\Sigma}_2^{-1}(\boldsymbol{x} - \boldsymbol{\mu}_2) - (\boldsymbol{x} - \boldsymbol{\mu}_1)^T\boldsymbol{\Sigma}_1^{-1}(\boldsymbol{x} - \boldsymbol{\mu}_1).$$

This completes the proof.

## C PROOF OF THEOREMS

### C.1 PROOF OF THEOREM 3.4

First, the convergence of $\widehat{\text{TV}}(\mathbb{P}, \mathbb{Q})$ to $\text{TV}(\mathbb{P}, \mathbb{Q})$ is implied by the convergence of $R(\widehat{f}) - R(f^\star)$, where $\widehat{f}$ be the plug-in classifier defined in (5). Specifically,

$$\widehat{f}(\boldsymbol{x}) = I\left(\phi(\widehat{h}(\boldsymbol{x})) > 1/2\right) = I\left(\frac{\exp(\widehat{h}(\boldsymbol{x}))}{1 + \exp(\widehat{h}(\boldsymbol{x}))} > 1/2\right).$$

To simplify notation, we denote $\widehat{\eta}(\boldsymbol{x}) = \phi(\widehat{h}(\boldsymbol{x}))$.

**Step 1: Establishing the connection between $R(\widehat{f}) - R(f^\star)$ and $\|\eta - \widehat{\eta}\|_{L_2(\mathbb{P}_{\boldsymbol{X}})}^2$**

Specifically, we first decompose $R(\widehat{f}) - R(f^\star)$ into two parts:

$$R(\widehat{f}) - R(f^\star) = \mathbb{E}\left[I(\widehat{f}(\boldsymbol{X}) \neq f^\star(\boldsymbol{X}))|2\eta(\boldsymbol{X}) - 1|\right]$$
$$= 2\mathbb{E}\left[I(\widehat{f}(\boldsymbol{X}) \neq f^\star(\boldsymbol{X}))|\eta(\boldsymbol{X}) - 1/2|I(|\eta(\boldsymbol{X}) - 1/2| < t)\right]$$
$$+ 2\mathbb{E}\left[I(\widehat{f}(\boldsymbol{X}) \neq f^\star(\boldsymbol{X}))|\eta(\boldsymbol{X}) - 1/2|I(|\eta(\boldsymbol{X}) - 1/2| \geq t)\right] \triangleq I_1 + I_2,$$

for any positive constant $t > 0$.

Next, we turn to bound $I_1$ and $I_2$ separately. Following from the fact that $|\eta(\boldsymbol{x}) - 1/2| \leq |\eta(\boldsymbol{x}) - \widehat{\eta}(\boldsymbol{x})|$ when $\widehat{f}(\boldsymbol{x}) \neq f^\star(\boldsymbol{x})$, we have

$$I_1 = 2\mathbb{E}\left[I(\widehat{f}(\boldsymbol{X}) \neq f^\star(\boldsymbol{X}))|\eta(\boldsymbol{X}) - 1/2|I(|\eta(\boldsymbol{X}) - 1/2| < t)\right]$$
$$\leq 2\mathbb{E}\left[I(\widehat{f}(\boldsymbol{X}) \neq f^\star(\boldsymbol{X}))|\eta(\boldsymbol{X}) - \widehat{\eta}(\boldsymbol{x})|I(|\eta(\boldsymbol{X}) - 1/2| < t)\right]$$
$$\leq 2\sqrt{\mathbb{E}\left[(\eta(\boldsymbol{X}) - \widehat{\eta}(\boldsymbol{x}))^2\right]} \cdot \sqrt{\mathbb{P}(|\eta(\boldsymbol{X}) - 1/2| < t)} \leq 2\|\eta - \widehat{\eta}\|_{L_2(\mathbb{P}_{\boldsymbol{X}})}C_0^{1/2}t^{\gamma/2}, \tag{10}$$

where the last inequality follows from the Cauchy–Schwarz inequality.

Next, $I_2$ can be bounded as

$$
\begin{aligned}
I_2 =& 2\mathbb{E}\Big[I(\widehat{f}(\boldsymbol{X}) \neq f^\star(\boldsymbol{X}))|\eta(\boldsymbol{X}) - 1/2|I(|\eta(\boldsymbol{X}) - 1/2| \geq t)\Big] \\
\leq& 2\mathbb{E}\Big[I(\widehat{f}(\boldsymbol{X}) \neq f^\star(\boldsymbol{X}))|\eta(\boldsymbol{X}) - \widehat{\eta}(\boldsymbol{x})|I(|\eta(\boldsymbol{X}) - 1/2| \geq t)\Big] \\
\leq& 2\mathbb{E}\Big[(\eta(\boldsymbol{X}) - \widehat{\eta}(\boldsymbol{x}))^2\Big]t^{-1} = 2t^{-1}\|\eta - \widehat{\eta}\|_{L_2(\mathbb{P}_{\boldsymbol{X}})}^2.
\end{aligned}
\tag{11}
$$

Combining (10) and (11) yields

$$
R(\widehat{f}) - R(f^\star) \leq 2\|\eta - \widehat{\eta}\|_{L_2(\mathbb{P}_{\boldsymbol{X}})}C_0^{1/2}t^{\gamma/2} + 2t^{-1}\|\eta - \widehat{\eta}\|_{L_2(\mathbb{P}_{\boldsymbol{X}})}^2.
$$

Setting $t = C_0^{-\frac{1}{\gamma+2}}\|\eta - \widehat{\eta}\|_{L_2(\mathbb{P}_{\boldsymbol{X}})}^{\frac{2}{\gamma+2}}$ yields

$$
R(\widehat{f}) - R(f^\star) \leq 4C_0^{\frac{1}{\gamma+2}}\left(\|\eta - \widehat{\eta}\|_{L_2(\mathbb{P}_{\boldsymbol{X}})}^2\right)^{\frac{\gamma+1}{\gamma+2}}.
$$

**Step 2. Establish the convergence of $\|\eta - \widehat{\eta}\|_{L_2(\mathbb{P}_{\boldsymbol{X}})}^2$**

For the mixed dataset $\mathcal{D} = \{\boldsymbol{x}_i\}_{i=1}^n \cup \{\widetilde{\boldsymbol{x}}_i\}_{i=1}^n$, we introduce a dataset $\mathcal{D}_0 = \{(\boldsymbol{x}_i^{(0)}, y_i^{(0)})\}_{i=1}^{2n}$ with $(\boldsymbol{x}_i^{(0)}, y_i^{(0)}) = (\boldsymbol{x}_i, 1)$ and $(\boldsymbol{x}_{n+i}^{(0)}, y_{n+i}^{(0)}) = (\widetilde{\boldsymbol{x}}_i, 0)$. Here $\mathcal{D}_0$ can be understood as a set of i.i.d. realizations of $(\boldsymbol{X}, Y)$ with $\boldsymbol{X} \sim \frac{1}{2}N(\boldsymbol{\mu}_1, \boldsymbol{\Sigma}_1) + \frac{1}{2}N(\boldsymbol{\mu}_2, \boldsymbol{\Sigma}_2)$ and $P(Y = 1|\boldsymbol{X}) = \frac{\mathbb{P}(\boldsymbol{X})}{\mathbb{P}(\boldsymbol{X}) + \mathbb{Q}(\boldsymbol{X})}$. Under the distribution of $(\boldsymbol{X}, Y)$, we first define $R_\phi(h) = \mathbb{E}\left[(\phi(h(\boldsymbol{X})) - Y)^2\right]$ as

$$
\begin{aligned}
R_\phi(h) - R_\phi(h_\phi^\star) =& \mathbb{E}\left[(\phi(h(\boldsymbol{X})) - Y)^2\right] - \mathbb{E}\left[(\phi(h_\phi^\star(\boldsymbol{X})) - Y)^2\right] \\
=& \mathbb{E}\left\{\eta(\boldsymbol{X})[\phi(h(\boldsymbol{X})) - 1]^2 + [1 - \eta(\boldsymbol{X})]\phi^2(h(\boldsymbol{X})) - \eta(\boldsymbol{X})(1 - \eta(\boldsymbol{X}))\right\} \\
=& \mathbb{E}\left[(\eta(\boldsymbol{X}) - \phi(h(\boldsymbol{X}))^2\right] = \|\eta - \eta_h\|_{L_2(\mathbb{P}_{\boldsymbol{X}})}^2.
\end{aligned}
$$

Next, we define $\widehat{R}_\phi(h)$ as an empirical version of $R_\phi(h)$.

$$
\widehat{R}_\phi(h) = \frac{1}{2n}\sum_{i=1}^{2n}\left(\phi(h(\boldsymbol{x}_i^{(0)})) - y_i^{(0)}\right)^2.
$$

Here $\widehat{h} = \arg\min_{h \in \mathcal{H}} \widehat{R}_\phi(h) + \lambda\|\boldsymbol{\beta}\|_2^2$ and $\widehat{\eta}(\boldsymbol{x}) = \phi(\widehat{h}(\boldsymbol{x}))$. Denote $\mathcal{A} = \{\mathcal{D} : \|\eta - \widehat{\eta}\|_{L_2(\mathbb{P}_{\boldsymbol{X}})}^2 > \delta\}$ and let $\mathcal{H}_0 = \{h \in \mathcal{H} : \|\eta - \eta_h\|_{L_2(\mathbb{P}_{\boldsymbol{X}})}^2 > \delta\}$ be a subset of the function class $\mathcal{H}$. First, if the dataset $\mathcal{D}_0 \in \mathcal{A}$, then we have $\widehat{h} \in \mathcal{H}_0$, implying $\sup_{h \in \mathcal{H}_0} \widehat{R}_\phi(h_\phi^\star) - \widehat{R}_\phi(h) + \lambda(\|\boldsymbol{\beta}^\star\|_2^2 - \|\boldsymbol{\beta}\|_2^2) \geq 0$ due to the optimality of $\widehat{h}$ in minimizing $\widehat{R}_\phi(h)$ within $\mathcal{H}$. Therefore, we have

$$
P(\mathcal{A}) \leq P\left(\sup_{f \in \mathcal{H}_0} \widehat{R}_\phi(h_\phi^\star) - \widehat{R}(h) + \lambda\|\boldsymbol{\beta}^\star\|_2^2 - \lambda\|\boldsymbol{\beta}\|_2^2 \geq 0\right).
\tag{12}
$$

Next we can decompose $\mathcal{H}_0$ as $\mathcal{H}_0 = \cup_{i=1}^\infty \mathcal{H}_0^{(i)}$ with $\mathcal{H}_0^{(i)}$ being defined as

$$
\mathcal{H}_0^{(i)} = \{h \in \mathcal{H}_0 : 2^{i-1}\delta \leq R_\phi(h) - R_\phi(h_\phi^\star) \leq 2^i\delta\}
$$

Therefore, (12) can be equivalently written as

$$
\begin{aligned}
P(\mathcal{A}) \leq & P\left(\sup_{\cup_{i=1}^{\infty} \mathcal{H}_0^{(i)}} \widehat{R}_\phi(h_\phi^\star) - \widehat{R}(h) + \lambda\|\boldsymbol{\beta}^\star\|_2^2 - \lambda\|\boldsymbol{\beta}\|_2^2 \geq 0\right) \\
\leq & \sum_{i=1}^{\infty} P\left(\sup_{h \in \mathcal{H}_0^{(i)}} \widehat{R}_\phi(h^\star) - \widehat{R}_\phi(h) + \lambda\|\boldsymbol{\beta}^\star\|_2^2 - \lambda\|\boldsymbol{\beta}\|_2^2 > 0\right) \\
\leq & \sum_{i=1}^{\infty} P\left(\sup_{h \in \mathcal{H}_0^{(i)}} \widehat{R}_\phi(h_\phi^\star) - R_\phi(f_\phi^\star) - \widehat{R}_\phi(h) + R_\phi(h) > \inf_{h \in \mathcal{H}_0^{(i)}} R_\phi(h) - R_\phi(h_\phi^\star) + \lambda\|\boldsymbol{\beta}\|_2^2 - \lambda\|\boldsymbol{\beta}^\star\|_2^2\right) \\
\leq & \sum_{i=1}^{\infty} P\left(\sup_{h \in \mathcal{H}_0^{(i)}} \widehat{R}_\phi(h_\phi^\star) - R_\phi(h_\phi^\star) - \widehat{R}_\phi(h) + R_\phi(h) > 2^{i-1}\delta - \lambda\|\boldsymbol{\beta}^\star\|_2^2\right) \\
\leq & \sum_{i=1}^{\infty} P\left(\sup_{h \in \mathcal{H}_0^{(i)}} \widehat{R}_\phi(h_\phi^\star) - R_\phi(h_\phi^\star) - \widehat{R}_\phi(h) + R_\phi(h) > 2^{i-2}\delta - \lambda\|\boldsymbol{\beta}^\star\|_2^2\right) \triangleq \sum_{i=1}^{\infty} I_i.
\end{aligned}
$$

where the last inequality by choosing $\lambda = \delta/(2\|\boldsymbol{\beta}^\star\|_2^2)$.

**Step 3. Bounding $I_i$**

First, we define

$$
\begin{aligned}
D_i(h) &= \left(\phi(h_\phi^\star(\boldsymbol{x}_i^{(0)})) - y_i^{(0)}\right)^2 - \left(\phi(h(\boldsymbol{x}_i^{(0)})) - y_i^{(0)}\right)^2, \\
D(h) &= \mathbb{E}\left[(\phi(h_\phi^\star(\boldsymbol{X})) - Y)^2 - (\phi(h(\boldsymbol{X})) - Y)^2\right].
\end{aligned}
$$

Then $I_i$ can be rewritten as

$$
\begin{aligned}
I_i =& P\left(\sup_{h \in \mathcal{H}_0^{(i)}} \frac{1}{2n}\sum_{i=1}^{2n}[D_i(h) - D(h)] > 2^{i-2}\delta\right) \\
=& P\left(\sup_{h \in \mathcal{H}_0^{(i)}} \frac{1}{2n}\sum_{i=1}^{2n}[D_i(h) - D(h)] - \nu_i(\mathcal{D}_0) > 2^{i-2}\delta - \nu_i(\mathcal{D}_0)\right),
\end{aligned}
$$

where $\nu_i(\mathcal{D}_0) = \mathbb{E}\left[\sup_{h \in \mathcal{H}_0^{(i)}} \frac{1}{2n}\sum_{i=1}^{2n}[D_i(h) - D(h)]\right]$. Here we assume $\nu_i(\mathcal{D}_0) \leq 2^{i-3}\delta$ and then we have

$$
I_i \leq P\left(\sup_{h \in \mathcal{H}_0^{(i)}} \frac{1}{2n}\sum_{i=1}^{2n}[D_i(h) - D(h)] - \nu_i(\mathcal{D}_0) > 2^{i-3}\delta\right).
$$

**Step 4. Verifying $\nu_i(\mathcal{D}_0) \leq 2^{i-3}\delta$ for $i \geq 1$**

Next, we intend to present the conditions under which $\nu_i(\mathcal{D}_0) \leq 2^{i-2}\delta$. Let $\mathcal{D}'$ be an independent copy of $\mathcal{D}_0$ and $(\tau_i)_{i=1}^{2n}$ be independent Rademacher random variables. Then we have

$$
\nu_i(\mathcal{D}_0) = \frac{1}{2n}\mathbb{E}_{\mathcal{D}_0}\left(\sup_{h\in\mathcal{H}_0^{(i)}}\sum_{i=1}^{2n}[D_i(h) - D(h)]\right) \leq \frac{1}{2n}\mathbb{E}_{\mathcal{D}_0}\left\{\mathbb{E}_{\mathcal{D}'}\left(\sup_{h\in\mathcal{H}_0^{(i)}}\sum_{i=1}^{2n}[D_i(h) - D_i'(h)]\Big|\mathcal{D}_0\right)\right\}
$$

$$
= \frac{1}{2n}\mathbb{E}_{\mathcal{D}_0,\mathcal{D}'}\left(\sup_{h\in\mathcal{H}_0^{(i)}}\sum_{i=1}^{2n}\tau_i[D_i(h) - D_i'(h)]\right)
$$

$$
= \frac{1}{2n}\mathbb{E}_{\mathcal{D}_0,\mathcal{D}'}\left(\sup_{h\in\mathcal{H}_0^{(i)}}\sum_{i=1}^{2n}\tau_i[D_i(h) - D_i(h_0) + D_i'(h_0) - D_i'(h)]\right)
$$

$$
\leq \frac{1}{n}\mathbb{E}\left(\sup_{h\in\mathcal{H}_0^{(i)}}\sum_{i=1}^{2n}\tau_i[D_i(h) - D_i(h_0)]\right),
$$

for any $h_0 \in \mathcal{H}_0^{(i)}$. Here the first inequality follows from the Jensen's inequality, and the second equality follows from the standard symmetrization argument.

Note that conditional on $\mathcal{D}_0$, $\frac{1}{\sqrt{2n}}\sum_{i=1}^{2n}\tau_i D_i(h)$ is a sub-Gaussian process with respect to $d$, where

$$
\rho^2(h_1, h_2) = \frac{1}{2n}\sum_{i=1}^{2n}\left(D_i(h_1) - D_i(h_2)\right)^2,
$$

for any $h_1, h_2 \in \mathcal{H}_0^{(i)}$. It then follows from Theorem 3.1 of Koltchinskii (2011) that

$$
\frac{1}{\sqrt{2n}}\mathbb{E}_{\mathcal{D}_0}\left(\sup_{h\in\mathcal{H}_0^{(i)}}\sum_{i=1}^{2n}\tau_i[D_i(h) - D_i(h_0)]\right) \lesssim \mathbb{E}\left(\int_0^{D(\mathcal{H}_0^{(i)})} H^{1/2}(\mathcal{H}_0^{(i)},\rho,\eta)d\eta\right),
$$

where $D(\mathcal{H}_0^{(i)})$ is the diameter of $\mathcal{H}_0^{(i)}$ with respect to $\rho$, and $H(\mathcal{H}_0^{(i)},\rho,\eta)$ is the $\eta$-entropy of $(\mathcal{H}_0^{(i)},\rho)$. For any $h_1, h_2 \in \mathcal{H}_0^{(i)}$, it follows that

$$
\mathbb{E}\rho^2(h_1, h_2) = \frac{1}{2n}\sum_{i=1}^{2n}\mathbb{E}\left(D_i(h_1) - D_i(h_2)\right)^2 \leq \frac{1}{2n}\sum_{i=1}^{2n}\mathbb{E}\left(D_i^2(h_1)\right) + \frac{1}{2n}\sum_{i=1}^{2n}\mathbb{E}\left(D_i^2(h_2)\right)
$$

$$
\leq 8\|\eta_{h_1} - \eta\|_{L_2(\mathbb{P}_{\boldsymbol{X}})}^2 + 8\|\eta_{h_2} - \eta\|_{L_2(\mathbb{P}_{\boldsymbol{X}})}^2.
$$

Therefore, we get

$$
\mathbb{E}D(\mathcal{H}_0^{(i)}) \leq 4\sup_{h\in\mathcal{H}_0^{(i)}}\|\eta_h - \eta\|_{L_2(\mathbb{P}_{\boldsymbol{X}})} \leq \sqrt{2^{i+4}\delta}. \tag{13}
$$

Moreover,

$$
d^2(h_1, h_2) = \frac{1}{2n}\sum_{i=1}^{2n}\left(D_i(h_1) - D_i(h_2)\right)^2 = \frac{1}{2n}\sum_{i=1}^{2n}\left((\phi(h_1(\boldsymbol{x}_i^{(0)})) - y_i^{(0)})^2 - (\phi(h_2(\boldsymbol{x}_i^{(0)})) - y_i^{(0)})^2\right)^2
$$

$$
\leq \frac{2}{n}\sum_{i=1}^{2n}\left(\phi\left(h_1(\boldsymbol{x}_i^{(0)})\right) - \phi\left((h_2(\boldsymbol{x}_i^{(0)})\right)\right)^2 \leq \frac{1}{8n}\sum_{i=1}^{2n}\left(h_1(\boldsymbol{x}_i^{(0)}) - h_2(\boldsymbol{x}_i^{(0)})\right)^2
$$

$$
\leq \frac{1}{4}\|\boldsymbol{\beta}_1 - \boldsymbol{\beta}_2\|_2^2\frac{1}{2n}\sum_{i=1}^{2n}\|\psi(\boldsymbol{x}_i^{(0)})\|_2^2 \triangleq \frac{M(\mathcal{D}_0)}{4}\|\boldsymbol{\beta}_1 - \boldsymbol{\beta}_2\|_2^2
$$

where $h_1(\boldsymbol{x}) = \boldsymbol{\beta}_1^T\psi(\boldsymbol{x})$, $h_2(\boldsymbol{x}) = \boldsymbol{\beta}_2^T\psi(\boldsymbol{x})$, the second inequality follows from the fact that $\phi(x)$ is a 1/4-Lipschitz function, and $M(\mathcal{D}_0) = \frac{1}{2n}\sum_{i=1}^{2n}\|\psi(\boldsymbol{x}_i^{(0)})\|_2^2$. Thus, $\rho^2(h_1, h_2) \leq \eta^2$ if $\|\boldsymbol{\beta}_1 - \boldsymbol{\beta}_2\|_2^2 \leq \frac{M(\mathcal{D}_0)\eta^2}{4}$. This further leads to

$$
H(\mathcal{H}_0^{(i)},\rho,\eta) \leq H\left(B_2(d), \|\cdot\|_2, \frac{C_{\mathcal{H}}\sqrt{M(\mathcal{D}_0)}\eta}{2}\right) \leq d\log\left(\frac{6}{C_{\mathcal{H}}\sqrt{M(\mathcal{D}_0)}\eta}\right),
$$

where $B_2(d)$ is the unit $l_2$-ball in $\mathbb{R}^d$ and the last inequality follows by setting $\frac{6}{C_{\mathcal{H}}\sqrt{M(\mathcal{D}_0)\eta}} \leq 1$.

Then, applying the Dudley's integral entropy bound (Koltchinskii, 2011), we have

$$\nu_i(\mathcal{D}_0) \lesssim \frac{1}{\sqrt{n}}\mathbb{E}\left(\int_0^{D(\mathcal{H}_0^{(i)})} H^{1/2}\big(\mathcal{H}_0^{(i)}, d, \eta\big)d\eta\right)$$

$$\lesssim \mathbb{E}\left(\frac{1}{\sqrt{n}}\int_0^{D(\mathcal{H}_0^{(i)})} \sqrt{d\log\left(\frac{6}{C_{\mathcal{H}}\sqrt{M(\mathcal{D}_0)\eta}}\right)}d\eta\right).$$

For ease of notation, we let $C_1 = \frac{C_{\mathcal{H}}\sqrt{M(\mathcal{D}_0)}}{6}$. Next,

$$\sqrt{\frac{d}{n}}\int_0^{D(\mathcal{H}_0^{(i)})}\sqrt{\log\left(\frac{1}{C_1\eta}\right)}d\eta = \frac{\sqrt{d}}{C_1\sqrt{n}}\int_0^{C_1 D(\mathcal{H}_0^{(i)})}\sqrt{\log\left(\frac{1}{\eta}\right)}d\eta$$

$$= \frac{\sqrt{d}}{C_1\sqrt{n}}\int_0^{C_1 D(\mathcal{H}_0^{(i)})}\sqrt{\log\left(\frac{1}{\eta}\right)}d\eta = \frac{\sqrt{d}}{C_1\sqrt{n}}\int_{\frac{1}{C_1 D(\mathcal{H}_0^{(i)})}}^{+\infty}\frac{\sqrt{\log(t)}}{t^2}dt$$

$$= \frac{1}{C_1}\sqrt{\frac{d}{n\log\left(\frac{1}{C_1 D(\mathcal{H}_0^{(i)})}\right)}}\int_{\frac{1}{C_1 D(\mathcal{H}_0^{(i)})}}^{+\infty}\frac{\log(t)}{t^2}dt. \tag{14}$$

By the fact that $\int_a^\infty \log(x)/x^2 dx = (\log(a)+1)/a$, we further have

$$(14) \lesssim \frac{\sqrt{d}D(\mathcal{H}_0^{(i)})}{\sqrt{n}}\sqrt{\log\left(\frac{1}{C_1 D(\mathcal{H}_0^{(i)})}\right)},$$

where the inequality follows from the fact that $1/x + x \leq 2x$ for $x \geq 1$. Next, by the fact that $f(x,y) = x\sqrt{\log(\frac{1}{xy})}$ is a concave function, we further have

$$\mathbb{E}\left(\frac{\sqrt{d}D(\mathcal{H}_0^{(i)})}{\sqrt{n}}\sqrt{\log\left(\frac{1}{C_1 D(\mathcal{H}_0^{(i)})}\right)}\right)$$

$$\leq \mathbb{E}\left(D(\mathcal{H}_0^{(i)})\right)\sqrt{\frac{d}{n}}\sqrt{\log\left(\frac{1}{\mathbb{E}\left(C_1 D(\mathcal{H}_0^{(i)})\right)}\right)} \lesssim \sqrt{\frac{d2^{i+4}\delta}{n}}\sqrt{\log\left(\frac{1}{2^{i+4}\delta}\right)}.$$

If $\sqrt{\frac{d2^{i+4}\delta}{n}}\sqrt{\log\left(\frac{1}{2^{i+4}\delta}\right)} \leq 2^i\delta$, we have $\frac{d}{n}\log(n/d) \lesssim \delta$.

**Step 5. Bounding $\sum_{i=1}^\infty I_i$**

Applying Theorem 1.1 of (Klein & Rio, 2005) to $I_i$, we have

$$I_1 \leq \exp\left(-\frac{2^{2i-4}\delta^2 n^2}{8\nu_i(\mathcal{D}_0)n + 2V_i n + 3\cdot 2^{i-3}\delta n}\right) \leq \exp\left(-\frac{2^{2i-4}\delta^2 n^2}{8V_i n + 7\cdot 2^{i-2}\delta n}\right) \tag{15}$$

where $V_i = \sup_{h\in\mathcal{H}_0^{(i)}} \text{Var}\left[(\phi(h_\phi^\star(\boldsymbol{X})) - Y)^2 - (\phi(h(\boldsymbol{X})) - Y)^2\right]$. Next, we establish the relation betwen $V$ and $\delta$.

$$V_i = \sup_{h\in\mathcal{H}_0^{(i)}} \text{Var}\left[(\phi(h_\phi^\star(\boldsymbol{X})) - Y)^2 - (\phi(h(\boldsymbol{X})) - Y)^2\right]$$

$$\leq \sup_{h\in\mathcal{H}_0^{(i)}} \mathbb{E}\left[(\phi(h_\phi^\star(\boldsymbol{X})) - Y)^2 - (\phi(h(\boldsymbol{X})) - Y)^2\right]^2$$

$$\leq \sup_{h\in\mathcal{H}_0^{(i)}} \mathbb{E}\left[(\phi(h_\phi^\star(\boldsymbol{X})) - \phi(h(\boldsymbol{X}))) \cdot (\phi(h_\phi^\star(\boldsymbol{X})) + \phi(h(\boldsymbol{X})) - 2Y)\right]^2$$

$$\leq 4\sup_{h\in\mathcal{H}_0^{(i)}} \mathbb{E}\left[\phi(h_\phi^\star(\boldsymbol{X})) - \phi(h(\boldsymbol{X}))\right]^2 \leq 4\sup_{h\in\mathcal{H}_0^{(i)}} \|\eta - \eta_h\|_{L_2(\mathbb{P}_{\boldsymbol{X}})}^2 \leq 2^{i+2}\delta.$$

Therefore, (15) can be further bounded as $I_i \leq \exp\left(-C2^i \delta n\right) \leq \exp\left(-Ci\delta n\right)$ for some positive constant $C$.

$$\sum_{i=1}^{\infty} I_i \leq \sum_{i=1}^{\infty} \exp\left(-Ci\delta n\right) \leq \frac{\exp(-C\delta n)}{1 - \exp(-C\delta n)} \lesssim \exp(-C\delta n).$$

Since $\frac{d}{2n}\log(n) \lesssim \delta$, we further have $\sum_{i=1}^{\infty} I_i \lesssim n^{-C}$ for some positive constant $C$. Finally, we have

$$\mathbb{P}\left(R(\widehat{f}) - R(f^\star) \geq 4C_0^{\frac{1}{\gamma+2}}\left(\frac{d\log n}{2n}\right)^{\frac{\gamma+1}{\gamma+2}}\right) \leq \mathbb{P}\left(\|\eta - \widehat{\eta}\|_{L_2(\mathbb{P}_{\boldsymbol{X}})}^2 \geq \frac{d\log n}{2n}\right) \lesssim n^{-C},$$

for some positive constant $C$. Therefore

$$\mathbb{E}_{\mathcal{D}}\left\{\mathrm{TV}(\mathbb{P}, \mathbb{Q}) - \widehat{\mathrm{TV}}(\mathbb{P}, \mathbb{Q})\right\} \leq 2\mathbb{E}\left(R(\widehat{f}) - R(f^\star)\right) \lesssim C_0^{\frac{1}{\gamma+2}}\left(\frac{d\log n}{2n}\right)^{\frac{2}{3}}. \tag{16}$$

This completes the proof.

## C.2   PROOF OF THEOREM 3.6

By Lemma 3.5, we have

$$P(|\eta(\boldsymbol{X}) - 1/2| < t) \leq \frac{2t}{(1-2c)\sqrt{\pi}\|\boldsymbol{\mu}_1 - \boldsymbol{\mu}_2\|_{\boldsymbol{\Sigma}}},$$

for any $t < c$ with $c \leq 1/2$. Furthermore, the result in (16) holds with $t = C_0^{-\frac{1}{\gamma+2}}\|\eta - \widehat{\eta}\|_{L_2(\mathbb{P}_{\boldsymbol{X}})}^{\frac{2}{\gamma+2}}$, which converges to 0 in probability. Therefore, we can simply consider $c = 1/4$ and the choice of $t$ is asymptotically achievable. Therefore, in the case of $\boldsymbol{\Sigma}_1 = \boldsymbol{\Sigma}_2 = \boldsymbol{\Sigma}$, $C_0$ in Assumption 3.1 becomes $\frac{4}{\sqrt{\pi}\|\boldsymbol{\mu}_1 - \boldsymbol{\mu}_2\|_{\boldsymbol{\Sigma}}}$. It then follows that

$$\mathbb{E}_{\mathcal{D}}\left\{\mathrm{TV}(\mathbb{P}, \mathbb{Q}) - \widehat{\mathrm{TV}}(\mathbb{P}, \mathbb{Q})\right\} \lesssim \left(\frac{1}{\|\boldsymbol{\mu}_1 - \boldsymbol{\mu}_2\|_{\boldsymbol{\Sigma}}}\right)^{\frac{1}{3}}\left(\frac{d\log n}{2n}\right)^{\frac{2}{3}}.$$

This completes the proof.

## C.3   PROOF OF THEOREM 3.7

**Proof of Theorem 3.7**: Let $\mathbb{P}(\boldsymbol{x})$ and $\mathbb{Q}(\boldsymbol{x})$ be the density functions of two different random variables from the exponential family:

$$\mathbb{P}(\boldsymbol{x}) = h_1(\boldsymbol{x}) \cdot \exp\left[\boldsymbol{\eta}_1(\boldsymbol{\theta_1}) \cdot \boldsymbol{T}_1(\boldsymbol{x}) - A_1(\boldsymbol{\theta_1})\right],$$
$$\mathbb{Q}(\boldsymbol{x}) = h_2(\boldsymbol{x}) \cdot \exp\left[\boldsymbol{\eta}_2(\boldsymbol{\theta_2}) \cdot \boldsymbol{T}_2(\boldsymbol{x}) - A_2(\boldsymbol{\theta_2})\right].$$

According to proof of Lemma 3.2, the optimal classifier is

$$f^\star(x) = \mathrm{sign}\left(\mathbb{P}(\boldsymbol{x}) - \mathbb{Q}(\boldsymbol{x})\right).$$

Observing that

$$\frac{\mathbb{P}(\boldsymbol{x})}{\mathbb{Q}(\boldsymbol{x})} = \frac{h_1(\boldsymbol{x}) \cdot \exp\left[\boldsymbol{\eta}_1(\boldsymbol{\theta_1}) \cdot \boldsymbol{T}_1(\boldsymbol{x}) - A_1(\boldsymbol{\theta_1})\right]}{h_2(\boldsymbol{x}) \cdot \exp\left[\boldsymbol{\eta}_2(\boldsymbol{\theta_2}) \cdot \boldsymbol{T}_2(\boldsymbol{x}) - A_2(\boldsymbol{\theta_2})\right]}$$
$$= \frac{h_1(\boldsymbol{x})}{h_2(\boldsymbol{x})} \cdot \exp\left[A_2(\boldsymbol{\theta_2}) - A_1(\boldsymbol{\theta_1}) + \boldsymbol{\eta}_1(\boldsymbol{\theta_1}) \cdot \boldsymbol{T}_1(\boldsymbol{x}) - \boldsymbol{\eta}_1(\boldsymbol{\theta_2}) \cdot \boldsymbol{T}_2(\boldsymbol{x})\right],$$

and that $\mathrm{sign}(\mathbb{P}(\boldsymbol{x}) - \mathbb{Q}(\boldsymbol{x})) = \mathrm{sign}\left(\log\left(\frac{\mathbb{P}(\boldsymbol{x})}{\mathbb{Q}(\boldsymbol{x})}\right)\right)$, thus the optimal classifier is given as

$$f^\star(x) = I\left(\log\left(\frac{h_1(\boldsymbol{x})}{h_2(\boldsymbol{x})}\right) + A_2(\boldsymbol{\theta_2}) - A_1(\boldsymbol{\theta_1}) + \boldsymbol{T}_1(\boldsymbol{x})\boldsymbol{\eta}_1(\boldsymbol{\theta_1}) - \boldsymbol{T}_2(\boldsymbol{x})\boldsymbol{\eta}(\boldsymbol{\theta_2}) > 0\right).$$

This completes the proof.

Table 6: Competing Total Variation Estimation Methods

| Methods | Estimator | Samples |
|---|---|---|
| True Total Variation | $\frac{1}{N}\sum_{i=1}^{N}\left\|\frac{\mathbb{Q}(\boldsymbol{x}'_i)-\mathbb{P}(\boldsymbol{x}'_i)}{\mathbb{P}(\boldsymbol{x}'_i)+\mathbb{Q}(\boldsymbol{x}'_i)}\right\|$ | $\{\boldsymbol{x}'_i\}_{i=1}^{N} \overset{\text{i.i.d.}}{\sim} \frac{\mathbb{P}+\mathbb{Q}}{2}$ |
| Parameter Estimation | $\frac{1}{N}\sum_{i=1}^{N}\left\|\frac{\widehat{\mathbb{Q}}(\boldsymbol{x}'_i)-\widehat{\mathbb{P}}(\boldsymbol{x}'_i)}{\widehat{\mathbb{P}}(\boldsymbol{x}'_i)+\widehat{\mathbb{Q}}(\boldsymbol{x}'_i)}\right\|$ | $\{\boldsymbol{x}'_i\}_{i=1}^{N} \overset{\text{i.i.d.}}{\sim} \frac{\widehat{\mathbb{P}}+\widehat{\mathbb{Q}}}{2}$ |
| Kernel Density Estimation (Sasaki et al., 2015) | $\frac{1}{N}\sum_{i=1}^{N}\left\|\frac{\widetilde{\mathbb{Q}}_{\text{kde}}(\boldsymbol{x}'_i)-\widetilde{\mathbb{P}}_{\text{kde}}(\boldsymbol{x}'_i)}{\widetilde{\mathbb{P}}_{\text{kde}}(\boldsymbol{x}'_i)+\widetilde{\mathbb{Q}}_{\text{kde}}(\boldsymbol{x}'_i)}\right\|$ | $\{\boldsymbol{x}'_i\}_{i=1}^{N} \overset{\text{i.i.d.}}{\sim} \frac{\widetilde{\mathbb{P}}_{\text{kde}}+\widetilde{\mathbb{Q}}_{\text{kde}}}{2}$ |
| Nearest Neighbor Ratio Estimation (Noshad et al., 2017) | $\frac{1}{n}\sum_{i=1}^{M}\frac{1}{2}\left\|\frac{\eta N_i}{M_i+1}-1\right\|$ | $\{\boldsymbol{x}'_i\}_{i=1}^{N} \overset{\text{i.i.d.}}{\sim} \mathbb{P}, \{\widetilde{\boldsymbol{x}}'_i\}_{i=1}^{M} \overset{\text{i.i.d.}}{\sim} \mathbb{Q}$ |
| Esemble Estimation (Moon & Hero, 2014) | $\frac{1}{n}\sum_{i=1}^{N}\frac{1}{2}\left\|\frac{M_2(\rho_{2,k}(i))^p}{M_1(\rho_{1,k}(i))^p}-1\right\|$ | $\{\boldsymbol{x}'_i\}_{i=1}^{M_1} \overset{\text{i.i.d.}}{\sim} \mathbb{P}, \{\widetilde{\boldsymbol{x}}'_i\}_{i=1}^{n} \overset{\text{i.i.d.}}{\sim} \mathbb{Q}$ |

## D EXPERIMENTAL SETTING

### D.1 SIMULATION STUDY

Two groups of Gaussian mixture ($\mathbb{D}(\boldsymbol{x}) = \frac{\mathbb{P}(\boldsymbol{x})+\mathbb{Q}(\boldsymbol{x})}{2}$) samples are generated as training data and testing data respectively. Training data size is set to either 1000 or 10,000, while test data size is 50,000.

**Discriminative estimation (DisE):** A classifier in the corresponding function class $\widehat{f}$ is trained with the use of transformed training data, and its classification error in test data can be used to estimate total variation via $\widehat{\text{TV}}(\mathbb{P},\mathbb{Q}) = 1 - 2R(\widehat{f})$.

**True total variation:** Since there is no closed form of TV distance between two Gaussian distributions, as the standard, we employ the Monte Carlo method to approximate the true total variation via $\text{TV}(\mathbb{P},\mathbb{Q}) \approx \frac{1}{N}\sum_{i=1}^{N}\left|\frac{\mathbb{Q}(\boldsymbol{x}'_i)-\mathbb{P}(\boldsymbol{x}'_i)}{\mathbb{P}(\boldsymbol{x}'_i)+\mathbb{Q}(\boldsymbol{x}'_i)}\right|$, where $\{\boldsymbol{x}'_i\}_{i=1}^{N} \overset{\text{i.i.d.}}{\sim} \frac{\mathbb{P}+\mathbb{Q}}{2}$.

**Parameter estimation (PE):** $\text{TV}(\widehat{\mathbb{P}},\widehat{\mathbb{Q}}) \approx \frac{1}{N}\sum_{i=1}^{N}\left|\frac{\widehat{\mathbb{Q}}(\boldsymbol{x}'_i)-\widehat{\mathbb{P}}(\boldsymbol{x}'_i)}{\widehat{\mathbb{P}}(\boldsymbol{x}'_i)+\widehat{\mathbb{Q}}(\boldsymbol{x}'_i)}\right|$, $\widehat{\mathbb{P}}$ and $\widehat{\mathbb{Q}}$ denote the multivariate Gaussian distribution with parameters estimated based on $\{\boldsymbol{x}_i\}_{i=1}^{n}$ and $\{\widetilde{\boldsymbol{x}}_i\}_{i=1}^{n}$, respectively.

**Kernel density estimation (KDE):** $\text{TV}(\widetilde{\mathbb{P}},\widetilde{\mathbb{Q}}) \approx \frac{1}{N}\sum_{i=1}^{N}\left|\frac{\widetilde{\mathbb{Q}}(\boldsymbol{x}'_i)-\widetilde{\mathbb{P}}(\boldsymbol{x}'_i)}{\widetilde{\mathbb{P}}(\boldsymbol{x}'_i)+\widetilde{\mathbb{Q}}(\boldsymbol{x}'_i)}\right|$, where $\widetilde{\mathbb{P}}$ and $\widetilde{\mathbb{Q}}$ denote the kernel density estimation based on $\{\boldsymbol{x}_i\}_{i=1}^{n}$ and $\{\widetilde{\boldsymbol{x}}_i\}_{i=1}^{n}$, respectively. We select the optimal bandwidth based on Silverman's rule of thumb (Silverman, 2018).

**Nearest neighbor ratio estimation (NNRE):** $\text{TV}(\mathbb{P},\mathbb{Q}) \approx \frac{1}{M}\sum_{i=1}^{M}\tilde{g}\left(\frac{\eta N_i}{M_i+1}\right)$, where $\tilde{g}(x) = \frac{1}{2}|x-1|$, $\eta = \frac{M}{N}$ is ratio of samples from $\mathbb{P}$ and $\mathbb{Q}$. For each sample $\boldsymbol{x}'_i$ from $\{\widetilde{\boldsymbol{x}}_i\}_{i=1}^{M}$, find out the $k$ nearest neighbors in $\{\boldsymbol{x}_i\}_{i=1}^{N} \cup \{\widetilde{\boldsymbol{x}}_i\}_{i=1}^{M}$, among which $N_i$ points from $\{\boldsymbol{x}_i\}_{i=1}^{N}$ and $M_i$ points from $\{\widetilde{\boldsymbol{x}}_i\}_{i=1}^{M}$. We select the optimal choice of $k = \sqrt{M}$ (Noshad et al., 2017)

**Ensemble estimation (EE):** $\text{TV}(\mathbb{P},\mathbb{Q}) \approx \frac{1}{n}\sum_{i=1}^{N}\tilde{g}\left(\frac{M_2(\rho_{2,k}(i))^p}{M_1(\rho_{1,k}(i))^p}\right)$, where $\tilde{g}(x) = \frac{1}{2}|x-1|$. All samples in $\{\widetilde{\boldsymbol{x}}_i\}_{i=1}^{n}$ are divided into two sets $\{\widetilde{\boldsymbol{x}}_i\}_{i=1}^{N}$ and $\{\widetilde{\boldsymbol{x}}_i\}_{i=N+1}^{N+M_2}$, and $M_1$ samples are drawn from $\{\boldsymbol{x}_i\}_{i=1}^{n}$. For each sample $\boldsymbol{x}'_i$ from $\{\widetilde{\boldsymbol{x}}_i\}_{i=1}^{N}$, find out the distance of $k$- nearest neighbor of $\boldsymbol{x}'_i$ in $\{\boldsymbol{x}_i\}_{i=1}^{M_1}$, denoted by $\rho_{1,k}(i)$, and the distance of $k$- nearest neighbor of $\boldsymbol{x}'_i$ in $\{\widetilde{\boldsymbol{x}}_i\}_{i=N+1}^{n}$, denoted by $\rho_{2,k}(i)$. The optimal choice of $k = \sqrt{N}$ (Moon & Hero, 2014).

### D.2 REAL APPLICATION

With the use of MNIST and CIFAR-10 dataset, we train Generative Adversarial Network models for 100, 300, and 500 epochs, subsequently generating images with each of these models.

**MNIST database:** This dataset contains 70,000 grayscale images of handwritten digits. The dataset is divided into a training set of 60,000 images and a test set of 10,000 images. Each image is $28 \times 28$

pixels in size, and each pixel value ranges from 0 to 255, representing the intensity of the pixel. The dataset includes ten classes, corresponding to the digits 0 through 9.

**CIFAR-10 database:** This dataset contains $60,000$ color images of 10 distinct categories: airplanes, automobiles, birds, cats, deer, dogs, frogs, horses, ships, and trucks. The dataset is divided into a training set of $50,000$ images and a test set of $10,000$ images. Each color image is $32 \times 32$ pixels in size.

**Generative adversarial network models (GANs) settings for MNIST database:**

generator ={    Linear(100, 128),                discriminator ={    Linear(784, 256),
                 LeakyReLU(0.2, inplace=True),                                  LeakyReLU(0.2, inplace=True),
                 Linear(128, 256),                                        Linear(256, 128),
                 BatchNorm1d(256),                                        BatchNorm1d(256),
                 LeakyReLU(0.2, inplace=True),                                  LeakyReLU(0.2, inplace=True),
                 Linear(256, 784),                                        Linear(128, 1),
                 Tanh() }                                                  Sigmoid() }

Adam optimizer is used for both networks with learning rate 0.0002 and the loss function is defined to be binary cross-entropy loss.

**Generative adversarial network models (GANs) settings for CIFAR-10 database:**

generator ={    Linear(100, 2048),               discriminator ={    Conv2d(3, 64),
                 BatchNorm1d(2048),                                      LeakyReLU(0.2, inplace=True),
                 LeakyReLU(0.2, inplace=True),                                  Dropout(0.3),
                 ConvTranspose2d(512, 256),                            Conv2d(64, 128),
                 BatchNorm2d(256),                                      LeakyReLU(0.2, inplace=True),
                 LeakyReLU(0.2, inplace=True),                                  Dropout(0.3),
                 ConvTranspose2d(256, 128),                            Conv2d(128, 256),
                 BatchNorm2d(128),                                      LeakyReLU(0.2, inplace=True),
                 LeakyReLU(0.2, inplace=True),                                  Dropout(0.3),
                 ConvTranspose2d(128, 64),                            Conv2d(256, 512),
                 BatchNorm2d(64),                                        LeakyReLU(0.2, inplace=True),
                 LeakyReLU(0.2, inplace=True),                                  Dropout(0.3),
                 ConvTranspose2d(64, 3),                              Linear(20482, 1),
                 Tanh() }                                                Sigmoid() }

Each ConvTranspose2d layer and Conv2d layer are with these parameter settings: kernel size=5, stride=2, padding=2, output padding=1.

Then we use pretrained ResNet-18 model to find out the embedding of each image. We modify the output size in last fully-connected layer of this model to the desired dimension of embeddings {20, 35, 50}. After obtaining the embedding, we estimate TV distance between embedding of original images and generated images for each class using different approaches. Finally, we calculate mean values and standard deviation of all classes.

