# OpenReview forum: "Discriminative Estimation of Total Variation Distance: A Fidelity Auditor for Generative Data"
_ICLR.cc/2025/Conference — Submitted to ICLR 2025_

### Official Review · Reviewer_Zwqu · 2024-10-28

**Soundness:** 3
**Presentation:** 3
**Contribution:** 2
**Rating:** 5
**Confidence:** 3

**Summary:**

The paper shows how to efficiently compute the total variation (TV) distance between two distributions by deriving an equivalence between the TV distance and the Bayes-Optimal classifier. This immediately leads to a lower bound on the TV distance by choosing a non-optimal classifier that can be learned from data. The paper then proves that under a low noise condition, the lower bound for two Gaussians (which is efficiently computed) converges with a relatively rapid convergence rate. The experiments show that the proposed method outperforms existing methods for computing the TV distance.

**Strengths:**

The paper is clearly written and the derivation seems novel. According to [7]  the question of how to compute the TV between two Gaussians with different means is an open problem and this paper  presents a partial solution to the problem.

**Weaknesses:**

The main weakness is significance to a ML audience. The paper attempts to motivate the paper by appealing to the problem of estimating the fidelity of generative models but it is not at all clear that TV is an appropriate distance for such a task. In particular, (1)  modern generative models are not Gaussian distributions and (2) for a Gaussian distribution the estimation of other divergences (in particular the Wasserstein-2 distance and the KL divergence) can be computed in closed form.

Another weakness is that a more recent version of [7] claims that the open problem has now been solved (see footnote 1 in
https://arxiv.org/pdf/1810.08693). The solution appears in the appendix of:
https://arxiv.org/pdf/2303.04288

**Questions:**

Can you give an alternative motivation for using TV in ML?

How does your approximation differ from the tight bound of Arbas et al?

---

> ### Author Response · Authors · 2024-11-23
> **Response to Reviewer Zwqu**
>
> We are grateful to the reviewer for the thoughtful and detailed feedback. We have made our best effort to address your concerns and present our **point-by-point** response for your questions and weaknesses that you highlight below.
>
> **For Weakness:** We would like to share our understanding of this problem and highlight the motivation behind our paper. To our knowledge, any metric measuring the distributional difference between two datasets can be used to evaluate the quality of generative data, such as $f$-divergence, Wasserstein distance, and so on. However, there are several difficulties: (1) the high dimensionality of data and (2) unknown distribution forms. **The first problem** can be alleviated by employing an embedding model to obtain a low-dimensional representation of the data. For example, Inception V3 [1] takes $299\times 299\times 3$ images as input and outputs 1000-dimensional embeddings. To tackle the second challenge, a common approach is to assume the normality of embeddings, from which some metrics can have a closed-form formula, such as KL divergence and Wasserstein distance. Therefore, **two natural questions arise**
>
> - **Q1**: Is it possible to estimate other distributional metrics without a closed-form formula under the Gaussian assumption? Since these metrics also provide a perspective of data fidelity.
>
> - **Q2**: Is it possible to relax the Gaussian assumption?
>
> To answer the above two questions, we consider the estimation of TV distance under the Gaussian assumption (Theorems 3.4 and 3.5). Later, we extend our results to the general exponential family (Theorem 3.7). Our theoretical results provide partial answers to these two questions. The proposed framework can be extended to other members of $f$-divergence, depending on the choice of loss function used for evaluation. In what follows, we summarize some connections between $f$-divergence metrics and surrogate loss functions. For example,
>
> - Exponential loss and the Hellinger distance: $\min_{f}R_{\exp}(f)=\min_{f} \mathbb{E}(\exp(-f(X)Y))=1-2h^2(\mathbb{P},\mathbb{Q})$, where $X \sim \frac{\mathbb{P}(X)+\mathbb{Q}(X)}{2}$ and $h(\mathbb{P},\mathbb{Q})$ represents the Hellinger distance between the distributions  $\mathbb{P}$ and $\mathbb{Q}$.
>
> **Conclusion** As long as we can consistently estimate $\min_{f}R_{\exp}(f)$ or $\min_{f}R_{log}(f)$, it is achievable to estimate the associated divergence. Therefore, our current framework can be extended to other $f$-divergence metrics.
>
> **For Question 1:** As discussed above, the key motivation for estimating the TV distance comes from our Q1 above. The TV distance can be used to evaluate the distributional difference between real and synthetic data. However, no closed-form solution is provided, even under the Gaussian assumption. Additionally, **another motivation** for estimating the TV distance lies in **privacy auditing** within the domain of differential privacy [2]. Specifically, total variation distance is a fundamental measure in privacy analysis, directly tied to the operational guarantees of differential privacy. Specifically: (1) Quantifying Worst-Case Privacy Leakage: In $(\epsilon, \delta)$-differential privacy, TV distance measures the allowable divergence between the distributions of outputs generated from two neighboring datasets, with $\delta$ representing the probability of deviation beyond the $\epsilon$-bound. By providing a lower bound on total variation, our tool identifies the minimum possible divergence, effectively quantifying the worst-case privacy leakage that cannot be avoided by any mechanism under these conditions.  This is particularly relevant for applications such as generative data models, where synthetic data's resemblance to real data must be carefully balanced against strict privacy requirements.
>
>
>
> **For Question 2** We carefully reviewed the results in [3] and found that their analysis essentially provides \textbf{a range} for the TV distance between two Gaussians:
> $
> \frac{1}{200}\Delta \leq
> \text{TV}(N(\mu_1, \Sigma_1), N(\mu_2, \Sigma_2)) \leq \frac{1}{\sqrt{2}}\Delta
> $
> where $\Delta$ is deterministic, given $\mu_1, \mu_2, \Sigma_1$, and $\Sigma_2$. In contrast, our proposed method provides a consistent estimation of the TV distance. We also thank the reviewer for bringing this related literature to our attention, which we have now included in our discussion.
>
> **References**:
>
> [1] Szegedy, C., Vanhoucke, V., Ioffe, S., Shlens, J. and Wojna, Z., 2016. Rethinking the inception architecture for computer vision. In Proceedings of the IEEE conference on computer vision and pattern recognition (pp. 2818-2826).
>
> [2] Koskela, A. and Mohammadi, J., 2024. Black Box Differential Privacy Auditing Using Total Variation Distance. arXiv preprint arXiv:2406.04827.
>
> [3] Arbas, J., Ashtiani, H. and Liaw, C., 2023, July. Polynomial time and private learning of unbounded gaussian mixture models. In International Conference on Machine Learning (pp. 1018-1040). PMLR.

---

### Official Review · Reviewer_CtCY · 2024-11-03

**Soundness:** 2
**Presentation:** 3
**Contribution:** 2
**Rating:** 5
**Confidence:** 4

**Summary:**

This paper explores the evaluation of generative AI models using the total variation (TV) distance between the distributions of real and synthetic data. The proposed method utilizes the variational representation of the TV distance, which necessitates finding the Bayes classifier. The analytical results are based on the assumption that both distributions are multivariate Gaussian, as specified in Assumption 3.1. The proxy optimal classifier is determined by solving Equation (4), and Theorem 3.4 establishes a bound between the true and estimated values of the TV distances. The paper also presents numerical results using synthetic Gaussian distributions and the MNIST dataset to demonstrate the feasibility of estimating TV distance for the evaluation of generative models.

**Strengths:**

1- The paper investigates the ranking and evaluation of generative models using total variation distance, which is an interesting approach.

2- Although I have some comments on the theorems in the paper, the theoretical result in Theorem 3.4 on estimating TV distance from samples looks interesting.

**Weaknesses:**

1- The authors’ approach of using total variation distance to evaluate generated samples is restricted to Gaussian and exponential family models. This assumption might not hold for the distribution of generated data. Even when the samples are processed through an embedding model, the resulting distribution will probably have multiple modes due to the variety of samples and cannot be represented as a unimodal Gaussian or exponential family model. Therefore, the authors’ method may not be capable of accurately estimating the underlying TV distance and can only provide a lower-bound which may be loose in practice.

2- The literature review in the introduction (the second and third paragraphs) misses several existing frameworks for evaluating generative models that are based on established distance measures between probability distributions. The maximum mean discrepancy (MMD), an established distance, has been used in evaluating generative models, and the Kernel Inception Distance is a metric that estimates the MMD between two distributions. In addition, the discriminative approach applied to the Wasserstein distances has also been used in the literature to evaluate generative models. Another note, the FID (Wasserstein distance between Gaussians fitted to real and fake distributions) also represents the variational representation of the Wasserstein distance where the function set in the dual optimization is the set of quadratic functions. These relevant ideas have not been acknowledged in the introduction.

3- Using Assumption 3.1 in Theorem 3.4 does not make much sense, because the authors have already supposed the Gaussian distributions P and Q. Therefore, constants $C_0$ and $\gamma$ should follow from the Gaussian distribution parameters $\mu_1,\mu_2,\Sigma_1, \Sigma_2$. The theorem is supposed to bound these constants $C_0$ and $\gamma$ in terms of the distribution parameters, and should not define them as two general constants without any further analysis.

4- Theorem 3.6 (and also Lemma 3.2) seems an obvious statement, because the Bayes classifier of two distributions $P,Q$ is well-known to be the indicator of $P(x)/Q(x) \leq 1$.

5- The numerical experiments on real data are insufficient. MNIST is considered a relatively simple dataset. The study could benefit from extending these experiments to more complex datasets such as CelebA, LSUN, or ImageNet.

**Questions:**

1- Do the authors have numerical results on more complex image datasets than MNIST?

2- Can the authors bound the constants $C,\gamma$ in Assumption 3.1 for multivariate Gaussians in Theorem 3.4?

---

> ### Author Response · Authors · 2024-11-23
> **Response to Reviewer CtCY**
>
> We would like to express our gratitude to the reviewer for the insightful feedback. We have made every effort to address your concerns and present our **point-by-point** responses to the questions and weaknesses you highlighted below. All revisions are highlighted in Blue in the revised manucsript.
>
> **For Weakness 1:** We would like to share our understanding of this problem and highlight the motivation behind our paper. In the literature, distributional-difference metrics are used to evaluate the fidelity of generative data, such as $f$-divergence, Wasserstein distance, and so on. However, there are two main difficulties: (1) the high dimensionality of data and (2) unknown distribution forms. **The first problem** can be alleviated by employing an embedding model to obtain a low-dimensional representation of the data. For example, Inception V3 [1] takes $299\times 299\times 3$ images as input and outputs 1000-dimensional embeddings. To tackle the second challenge, a common approach is to assume the normality of embeddings, from which some metrics can have a closed-form formula, such as KL divergence and Wasserstein distance. However, **two natural questions arise**
>
> - **Q1**: Is it possible to estimate other distributional metrics without a closed-form formula under the Gaussian assumption? Since these metrics also provide a perspective of data fidelity.
>
> - **Q2**: Is it possible to relax the Gaussian assumption?
>
> To answer the above two questions, we consider the estimation of TV distance under the Gaussian assumption (Theorem 3.4). Later, we extend our results to the general exponential family (Theorem 3.6) for removing the Gaussian assumption. Our results provide partial answers to these two questions.
>
> **For Weakness 2:** We thank the reviewer for pointing out the shortcomings in our literature review. We have now included the methods you suggested in the updated version of our review.
>
>
> **For Weakness 3 and Question 2:** We thank the reviewer for pointing out the possible improvement in our method. **Following your suggestion**, we have specified those constants in Theorem 3.5. We specify the values of $C_0$ and $\gamma$ when $\mathbb{P}$ and $\mathbb{Q}$ are multivariate Gaussian distributions with identical covariance matrices. In this case, the result becomes
> $$
>  \mathbb{E}_{\mathcal{D}}
>   \Big(
> \widehat{\mathrm{TV}}(\mathbb{P}, \mathbb{Q})-\mathrm{TV}(\mathbb{P}, \mathbb{Q})
> \Big)
> \lesssim  \Delta^{-\frac{1}{3}} \Big(
> \frac{d\log n}{2n}
> \Big)^{\frac{2}{3}}
>  $$
> where $\Delta = ( \mathbf{\mu}_1-\mathbf{\mu}_2)^T \mathbf{\Sigma}( \mathbf{\mu}_1-\mathbf{\mu}_2) $. In this result, we show that $C_0 \asymp 1/ \Delta$ and $\gamma=1$. Our findings illustrate that the proposed discriminative estimation method achieves a rapid convergence rate of $O\Big(\Delta^{-1/3}n^{-\frac{2}{3}}\Big)$, accompanied by a logarithmic factor. Notably, as $\Delta$ tends towards infinity ($\mathbf{\mu}_1$ becomes **more different** from $\mathbf{\mu}_2$), the convergence rate **accelerates**, consistent with our second observation mentioned earlier.
>
> **For Weakness 5 and Question 1:** We thank the reviewer for the suggestion, which is also suggested by Reviewer jVgz. Following your suggestion, we further included the CIFAR10 dataset in our real application. Our experimental result shows that the proposed method ensures consistent fidelity ranking of image data.
>
> | Method | Embeddings|100 epochs |  300 epochs | 500 epochs | Correct Ranking |
> | --------| ----------- | ----------- | ----------- | ----------- |----------- |
> | DisE |Resnet18-20  |     0.332(0.031)| 0.274(0.035) |0.255(0.042)| Yes|
> | DisE  |Resnet18-35 |  0.463(0.038) | 0.378(0.055)  | 0.348(0.055) |Yes|
> | DisE  |Resnet18-50  | 0.577(0.041)  |0.483(0.059) | 0.444(0.038) |Yes|
>
>
> More details about the implementations can be found in the revised manuscript.

---

> > ### Comment · Reviewer_CtCY · 2024-11-30
> >
> > I thank the authors for their detailed responses to my comments. The response helps with my questions on the literature review and the bounds on constant $C_0,\gamma_0$ in the numerical evaluation. However, I am still not sure why the Wassesrstein distance or KL-divergence should be replaced by TV-disance in the evaluation? Does this have a practical motivation and are there settings where switching to TV distance offers a significant benefit over Wassesrstein distance? I think this question is not addressed in the work, can the authors explain more about the practical benefits of TV distance?

---

> > > ### Author Response · Authors · 2024-12-01
> > >
> > > Thank you for your response and comment. In our paper, we propose a discriminative method to estimate the TV-distance, which can work as a fidelity auditor to evaluate generative data fidelity.  The advantage of our method is that finding a classifier is easier compared to estimating other distance metrics, such as the Wasserstein distance and the Fréchet distance. Besides, our proposed discriminative approach can be extended to exponential family, not limited in Gaussian distribution. For example, consider the Wasserstein distance:
> > > $$
> > > W_p(\mathbb{P}, \mathbb{Q}) = \Big( \inf_{\gamma \in \Pi(\mathbb{P}, \mathbb{Q})} \int_{\mathcal{X} \times \mathcal{X}} d(x, y)^p \, d\gamma(x, y) \Big)^{1/p},
> > > $$
> > > where the infimum is taken over the set of joint distributions $\gamma$ whose marginals are $\mathbb{P}$ and $\mathbb{Q}$. Obtaining this infimum can be challenging in practice compared to finding a classifier.
> > >
> > > In the domain of computer vision, the FID score is the primary metric used to assess the quality of images generated by generative models, which is based on Wasserstein distance between multivariate Gaussians in the feature space. It is worth noting that the Fréchet Inception Distance (FID) not only provides an accurate ranking of the fidelity of generative data but also comes with significant computational demands. Specifically, the FID approach requires substantial memory resources due to the use of the Inception-V3 model for obtaining image embeddings.

---

### Official Review · Reviewer_jVgz · 2024-11-04

**Soundness:** 3
**Presentation:** 3
**Contribution:** 2
**Rating:** 5
**Confidence:** 2

**Summary:**

The paper addresses the challenge of assessing the fidelity of generative data. It introduces a discriminative method for estimating the total variation (TV) distance between two distributions, which serves as a measure of fidelity. The approach links the estimation of TV distance to the Bayes risk in classification. The authors present theoretical results on the convergence rate of estimation errors for TV distance between Gaussian distributions. It shows that estimation accuracy improves as the separation between the distributions increases. This relationship is supported by empirical simulations. The method is applied to rank the fidelity of synthetic image data using the MNIST dataset.

**Strengths:**

1.	Total variation is a well-established metric in statistics and can potentially provide a quantitative way to assess the fidelity of generative data.
2.	The authors link the estimation of TV distance to Bayes risk in classification and provides an effective estimation method based on finite samples.
3.	The paper shows a fast convergence rate when the two Gaussian distributions are well-separated.

**Weaknesses:**

1.	It is not clear why focusing on using TV instead of other metrics like Wasserstein distance for evaluating the fidelity of the generated data. For two Gaussians, Wasserstein distance has closed form solution. The paper only listed $f$-divergence in the related work.
2.	The experimental results are very limited. Is it possible to add some natural image data results, such as CIFAR 10?
3.	It is not clear how to effectively select the appropriate hypothesis class and classifier in practice to achieve accurate TV distance estimation.
4.	On page 3 below eq. (3), the authors mentioned that “Intuitively, if none of classifies yields  a large lower bound, then the synthetic data can be considered similar to the real data, indicating that their total variation distance is small.” I feel that it is not true, since each classifier only offers a lower bound. Even if all lower bounds are small, it doesn’t guarantee that the TV is small.

**Questions:**

Could the authors provide more experimental results in real application? For example, using CIFAR10.

---

> ### Author Response · Authors · 2024-11-23
> **Response to Reviewer jVgz**
>
> Thank you very much for the time and effort that you have put into reviewing our paper. We greatly appreciate your detailed comments and constructive suggestions for improving the quality of this paper. In what follows, we provide **point-by-point response** to the question and weaknesses that you highlight. All revisions are highlighted in Blue in the revised manuscript.
>
> **For weakness 1**: Basically, the proposed framework can be easily extended to other members of $f$-divergence, depending on the choice of loss function used for evaluation. In what follows, we summarize some connections between $f$-divergence metrics and surrogate loss functions:
>
> - Exponential loss and the Hellinger distance: $\min_{f}R_{\exp}(f)=\min_{f} \mathbb{E}(\exp(-f(X)Y))=1-2h^2(\mathbb{P},\mathbb{Q})$, where $X \sim \frac{\mathbb{P}(X)+\mathbb{Q}(X)}{2}$ and $h(\mathbb{P},\mathbb{Q})$ represents the Hellinger distance between the distributions  $\mathbb{P}$ and $\mathbb{Q}$.
>
> - Logistic loss and Jensen-Shannon divergence: $\min_{f}R_{log}(f)=\min_{f} \mathbb{E}(\log(1+\exp(-f(X)Y)))=\log 2 - 2JS(\mathbb{P},\mathbb{Q})$, where $X \sim \frac{\mathbb{P}(X)+\mathbb{Q}(X)}{2}$ and $JS(\mathbb{P},\mathbb{Q})$ represents the Jensen-Shannon divergence between the distributions  $\mathbb{P}$ and $\mathbb{Q}$.
>
> **The conclusion** is that as long as we can consistently estimate $\min_{f}R_{\exp}(f)$ or $\min_{f}R_{log}(f)$, it is achievable to estimate the associated divergence. Therefore, our current framework can be extended to other $f$-divergence metrics. More details about deriving these results can be found in [1]. Moreover, we agree that the Wasserstein distance is a practical metric under the Gaussian assumption. However, the **main purpose** of this paper is to provide a practical alternative for evaluating the fidelity of generative data. Furthermore, our results can be extended to other distributions belonging to the exponential family, as summarized in Table 1 of the manuscript.
>
>
> **For weakness 2 and Question** Following your suggestion, we have included CIFAR10 in our experiment for verifying the effectiveness of the proposed method. Our experimental result shows that the proposed method ensures consistent fidelity ranking of image data.
>
> | Method | Embeddings|100 epochs |  300 epochs | 500 epochs | Correct Ranking |
> | --------| ----------- | ----------- | ----------- | ----------- |----------- |
> | DisE |Resnet18-20  |     0.332(0.031)| 0.274(0.035) |0.255(0.042)| Yes|
> | DisE  |Resnet18-35 |  0.463(0.038) | 0.378(0.055)  | 0.348(0.055) |Yes|
> | DisE  |Resnet18-50  | 0.577(0.041)  |0.483(0.059) | 0.444(0.038) |Yes|
>
>
> **For weakness 3**  In this paper, our main focus is generalizing the Gaussian assumption to the exponential family. In Theorem 3.7, we specify the explicit form of optimal classifier for other exponential family. This result implies that the corresponding hypothesis class in our framework should be $\mathcal{H}=\{f(\mathbf{x})=\mathbf{\beta}^T \psi(\mathbf{x})\}$ with $\psi(\mathbf{x})=(T_1(\mathbf{x}),T_2(\mathbf{x}))$, where $T_1(\mathbf{x})$ and $T_2(\mathbf{x})$ are sufficient statistics of $\mathbb{P}$ and $\mathbb{Q}$. We also provide more explicit resuls for 1-dimensional case in Table 1 of our paper
>
> **For weakness 4** We thank the reviewer for carefully examining the claims in our paper. We apologize for the lack of clarity in our initial submission.
> $
> \text{TV}(\mathbb{P}, \mathbb{Q}) \ge 1-2R(\widehat{f})
> \triangleq
> \widehat{\text{TV}}(\mathbb{P}, \mathbb{Q})
> $
> Here, when $\widehat{f}=f^\star$ (The Bayes classifier), we have $\widehat{\text{TV}}(\mathbb{P}, \mathbb{Q})=\text{TV}(\mathbb{P}, \mathbb{Q})$.  Therefore, our claim is that if the Bayes classifier fails to discriminate between real and synthetic data, their underlying distributions should be the same. Following your suggestion, we have rewritten this claim to avoid any misunderstanding.
>
> **References**:
>
> [1] XuanLong Nguyen. Martin J. Wainwright. Michael I. Jordan. "On surrogate loss functions and f-divergences." Ann. Statist. 37 (2) 876 - 904

---

### Author Response · Authors · 2024-11-28

Dear Reviewers and Area Chairs,

We sincerely appreciate the time, efforts, and expertise you have devoted to reviewing our paper.  Your feedback has provided invaluable insights and guidance for enhancing the quality of our work. We have carefully addressed your questions and provided detailed responses to clarify your concerns about our research. We would be glad to have further discussion of your concerns and comments on our paper.

Once again, thank you for your dedication and support in helping us improve our research and we look forward to having more discussion with you.

---

### Meta-Review · Area_Chair_gYuj · 2024-12-17

**Metareview:**

This paper introduces a discriminative method to estimate the total variation (TV) distance between two distributions, linking it to the Bayes risk in classification. The method utilizes the variational representation of TV distance, approximated by learning a proxy Bayes-optimal classifier. Theoretical results establish a convergence bound for estimating TV distance between Gaussian distributions, showing improved accuracy as their separation increases. Empirical evaluations on synthetic Gaussian data and the MNIST dataset validate the method’s effectiveness in ranking fidelity of generative models. Results demonstrate that the proposed approach efficiently estimates TV distance and outperforms existing methods under low noise conditions.

The proposed TV-based approach has some merit from a theoretical perspective. However, as most reviewers point out, the motivation for employing TV is somewhat unclear. Moreover, it would be beneficial to include more real experimental results to demonstrate its efficacy in practice. Thus, the paper requires major revisions and is difficult to accept in its current form. I encourage the authors to address the reviewers' comments and resubmit the paper to a future venue.

**Additional Comments On Reviewer Discussion:**

Most of the authors wonder why total variation (TV) is used for the two Gaussian case, as the 2-Wasserstein distance or KL divergence can be computed in closed form. This issue was not well addressed in the rebuttal. A comparison with the 2-Wasserstein distance is necessary. Since the average score is 5, which is clearly below the borderline, I recommend rejecting the paper.

---

### Decision · Program_Chairs · 2025-01-22

Reject